

# Constraint of non-methane volatile organic compound emissions with TROPOMI HCHO observations and its impact on summertime surface ozone simulation over China

Shuzhuang Feng[1], Fei Jiang[1,2,5*], Tianlu Qian[3], Nan Wang[4], Mengwei Jia[1], Songci Zheng[1], Jiansong Chen[6], Fang Ying[6], Weimin Ju[1,2]

[1] *Jiangsu Provincial Key Laboratory of Geographic Information Science and Technology, International Institute for Earth System Science, Nanjing University, Nanjing, 210023, China*

[2] *Jiangsu Center for Collaborative Innovation in Geographical Information Resource Development and Application, Nanjing, 210023, China*

[3] *School of Geographic and Biologic Information, Nanjing University of Postsand Telecommunications, Nanjing, 210023, China*

[4] *College of Carbon Neutrality Future Technology, Sichuan University, Chengdu, 610207, China*

[5] *Frontiers Science Center for Critical Earth Material Cycling, Nanjing University, Nanjing, 210023, China*

[6] *Hangzhou Municipal Ecology and Environment Bureau, Hangzhou, 310020, China*

* Corresponding author: Tel.: +86-25-83597077; Fax: +86-25-83592288; E-mail address: jiangf@nju.edu.cn





## Abstract

Non-methane volatile organic compounds (NMVOC), serving as crucial precursors of $O_3$, have a significant impact on atmospheric oxidative capacity and $O_3$ formation. However, both anthropogenic and biogenic NMVOC emissions remain subject to considerable uncertainty. Here, we extended the Regional multi-Air Pollutant Assimilation System (RAPAS) with the EnKF algorithm to optimize NMVOC emissions in China by assimilating TROPOMI HCHO retrievals. We also simultaneously optimize $NO_x$ emissions by assimilating in-situ $NO_2$ observations to address the chemical feedback among VOC-$NO_x$-$O_3$. Furthermore, a process-based analysis was employed to quantify the impact of NMVOC emission changes on various chemical reactions related to $O_3$ formation and depletion. NMVOC emissions exhibited a substantial reduction of 50.2%, especially in forest-rich areas of central and southern China, revealing a prior overestimation of biogenic NMVOC emissions. The RAPAS significantly improved HCHO simulations, reducing biases by 75.7%, indicating a notable decrease in posterior emission uncertainties. Moreover, the posterior NMVOC emissions significantly corrected the prior overestimation in $O_3$ simulations, reducing biases by 49.3%. This can be primarily attributed to a significant decrease in the $RO_2 + NO$ reaction rate and an increase in the $NO_2 + OH$ reaction rate in the afternoon, thus limiting $O_3$ generation. Sensitivity analyses emphasized the necessity of considering both NMVOC and $NO_x$ emissions for a comprehensive assessment of $O_3$ chemistry. This study enhances our understanding of the effects of NMVOC emissions on $O_3$ production and can contribute to the development of effective emission reduction policies.

## Keywords

NMVOC emissions, $O_3$ pollution, Emission inversion, HCHO column retrievals, Data assimilation



## 1 Introduction

Since the Chinese government implemented the Air Pollution Prevention and Control Action Plan in 2013, there has been a notable reduction in $NO_x$ emissions (Zheng et al., 2018). However, despite these advancements, the issue of $O_3$ pollution persists and, in certain cases, has shown signs of worsening (Ren et al., 2022). The increase in $O_3$ concentration can be attributed not only to adverse meteorological conditions but also predominantly to unbalanced joint control of non-methane volatile organic compounds (NMVOCs) and nitrogen oxides ($NO_x$) (Li et al., 2020). NMVOCs are vital precursors of $O_3$ and have a substantial impact on the atmospheric oxidation capacity, thereby altering the lifetimes of other pollutants. Accurately quantifying NMVOC emissions holds significant importance in investigating their impact on $O_3$ chemistry and in formulating emission reduction policies.

Anthropogenic NMVOC emissions have traditionally been estimated using a "bottom-up" method. However, the accuracy and timeliness of these estimations face challenges owing to the scarcity of local measurements for emission factors, the incompleteness and unreliability of activity data, and the diverse range of species and technologies involved (Cao et al., 2018; Hong et al., 2017). Furthermore, uncertainties arise in model-ready NMVOC emissions due to spatial and temporal allocations using various "proxy" data for different source sectors (Li et al., 2017a). Li et al. (2021) reported substantial discrepancies among emission estimates in various studies, ranging 23% to 56%. Biogenic NMVOC emissions are typically estimated using models like the Model of Emissions of Gases and Aerosols from Nature (MEGAN) (Guenther et al., 2012) and the Biogenic Emission Inventory System (BEIS) (Pierce et al., 1998). NMVOC emissions result from the multiplication of plant-specific standard emission rates by dimensionless activity factors. Nonetheless, apart from inaccuracies in the distribution of plant functional types, empirical parameterization, especially concerning responses to temperature and drought stress, can introduce substantial uncertainties (Angot et al., 2020; Seco et al., 2022; Jiang et al., 2018). Warneke et al. (2010) determined isoprene emission rates through field measurements and conducted a comparison with MEGAN and BEIS estimates, revealing a notable tendency for MEGAN to overestimate emissions, while BEIS consistently underestimated them. Similarly, Marais et al. (2014) found that MEGAN's isoprene emission estimates were 5-8 times higher than the canopy-scale flux measurements obtained from African field campaigns.





A top-down approach, utilizing observed data, has been developed for estimating VOCs emissions. For instance, based on aircraft and ground-based field measurements, the source-receptor relationships algorithm with Lagrangian particle dispersion model (Fang et al., 2016), mixed layer gradient techniques (Mo et al., 2020), eddy covariance flux measurements (Yuan et al., 2015), and box model (Wang et al., 2020) have been employed to complement or verify bottom-up results. However, these approaches do not comprehensively consider the complex nonlinear chemical reactions and transport processes that VOCs undergo in the atmosphere. Formaldehyde (HCHO) and glyoxal (CHOCHO) in the atmosphere serve as crucial oxidization intermediates for various VOCs (Hong et al., 2021; Liu et al., 2012). Satellite-based observations can readily detect their presence in the form of vertical column density (VCD) from space, making them widely utilized for estimating NMVOC emissions. A commonly used approach assumes that the observed HCHO/CHOCHO columns are locally linearly correlated with VOC emission rates (Palmer et al., 2006; Liu et al., 2012). However, this approach does not consider the spatial offset resulting from chemistry reactions and transport processes. Chaliyakunnel et al. (2019) conducted a Bayesian analysis to derive an optimal estimate of VOC emissions using HCHO measurements over the Indian subcontinent. Their results indicated that biogenic VOC emissions modeled by MEGANv2.1 were overestimated by approximately 30–60%, whereas anthropogenic VOC emissions derived from the RETRO inventory were underestimated by 13–16%. Cao et al. (2018) employed the GEOS-Chem model and its adjoint, incorporating tropospheric HCHO and CHOCHO column data from the GOME-2A and OMI satellites as constraints, to quantify Chinese NMVOC emissions. They demonstrated a low bias in the MEGAN model, in contrast to the significant overestimation shown in Bauwens et al. (2016), especially in southern China.

Several investigations have been conducted to explore the implications of inverted VOC emissions on surface $O_3$. For instance, using the Eulerian box model, Zhou et al. (2023) employed concurrent VOC measurements to constrain anthropogenic VOC emissions. This led to improved simulations of VOCs and $O_3$, with a reduction in high emissions by 15%–36% in the Pearl River Delta (PRD) region. Local model biases in simulating the oxidation of NMVOCs and $O_3$ are closed related to uncertainties in $NO_x$ emissions (Wolfe et al., 2016; Chan Miller et al., 2017). To tackle these critical questions, Kaiser et al. (2018) applied an adjoint algorithm to estimate isoprene




emission over the southeast US by downwardly adjusting anthropogenic $NO_x$ emissions
by 50% to rectify $NO_2$ simulations. Their findings indicated that isoprene emissions
from MEGAN v2.1 were overestimated by an average of 40%, slightly lower than the
50% reduction in Bauwens et al. (2016). Souri et al. (2020) simultaneously optimized
NMVOC and $NO_x$ emissions utilizing OMPS-NM HCHO and OMI $NO_2$ retrievals in
East Asia. They found that predominantly anthropogenic NMVOC emissions from
MIX-Asia 2010 increased over the North China Plain (NCP), whereas predominantly
biogenic NMVOC emissions from MEGAN v2.1 decreased over southern China after
the adjustment. Unfortunately, the posterior simulations exacerbated the overestimation
of $O_3$ levels in northern China.
Most studies regarding the inversion of NMVOC emissions or its impact on $O_3$
neglected the uncertainties associated with $NO_x$-dependent production or loss of
NMVOC oxidation and $O_3$. An iteratively nonlinear joint inversion of $NO_x$ and
NMVOCs using multi-species observations is expected to minimize the uncertainties
in their emissions and is well-suited to address the intricate relationship among VOC-
$NO_x$-$O_3$. In this study, we extended the Regional multi-Air Pollutant Assimilation
System (RAPAS) upon the ensemble Kalman filter (EnKF) assimilation algorithm to
enhance the optimization of NMVOC emissions over China, utilizing the
TROPOspheric Monitoring Instrument (TROPOMI) HCHO retrievals with high spatial
coverage and resolution. To more accurately quantify the impact of NMVOC emissions
on $O_3$, $NO_x$ emissions were simultaneously adjusted using nationwide in-situ $NO_2$
observations. Process analysis was subsequently employed to quantify various
chemical pathways associated with $O_3$ formation and loss. Through a top-down
constraint on both emissions, this study aims to offer a more scientific insight into the
consequences of optimizing NMVOC emissions on $O_3$ and contribute to the
development of appropriate emission reduction policies.
**2 Data and Methods**
**2.1 Data Assimilation System**
The RAPAS system (Feng et al., 2023) has been developed based on a regional
chemical transport model (CTM) and ensemble square root filter (EnSRF) assimilation
modules (Whitaker and Hamill, 2002), which are employed for simulating atmospheric
compositions and inferring anthropogenic emissions by assimilating surface



observations, respectively (Feng et al., 2022; Feng et al., 2020). The inversion process
follows a two-step procedure within each inversion window. The two-step inversion
strategy facilitates error propagation and iterative emission optimization, which have
proven the superiority and robustness of our system in estimating emissions (Feng et
al., 2023). In this study, we extended the data frame to include the assimilation of
TROPOMI HCHO retrievals for optimizing NMVOC emissions. Concise descriptions
of the forecast model, data assimilation approach, and experimental settings follow.

### 2.1.1 Atmospheric Transport Model

The Weather Research and Forecast (WRF v4.0) model (Skamarock and Klemp, 2008)
and the Community Multiscale Air Quality Modeling System (CMAQ v5.0.2) (Byun
and Schere, 2006) were applied to simulate meteorological conditions and atmospheric
chemistry, respectively. WRF simulations were conducted with a 27-km horizontal
resolution, covering the entire mainland China on a grid of 225 × 165 cells (Figure 1).
The CMAQ model was run over the same domain, but with a removal of three grid cells
on each side of the WRF domain. The vertical settings in WRF and CMAQ was the
same as Feng et al. (2020). To account for the rapid expansion of urbanization, we
updated underlying surface information for urban and built-up land using the MODIS
Land Cover Type Product (MCD12C1) Version 6.1 of 2022. Chemical lateral boundary
conditions were extracted from the output of the global CTM Whole Atmosphere
Community Climate Model (WACCM) with a resolution of 0.9° × 1.25° at 6-hour
intervals (Marsh et al., 2013). In the first data assimilation (DA) window, chemical
initial conditions also originated from WACCM output, whereas in subsequent
windows, they were derived through forward simulation using optimized emissions
from the previous window. Table S1 lists the detailed physical and chemical
configurations. To assess the impact of updated NMVOC emissions on $O_3$ production
efficiency, we further decoupled the contribution of the primary chemical processes to
the $O_3$ levels using the CMAQ Integrated Reaction Rate (IRR) analysis.

### 2.1.2 EnKF Assimilation Algorithm

The emissions are constrained using the Ensemble Square Root Filter (EnSRF)
algorithm introduced by Whitaker and Hamill (2002). This approach fully accounts for
temporal and geographical variations in both the transportation and chemical reactions
within the emission estimates. During the forecast step, the background ensembles are



derived by applying perturbation to the prior emissions. The perturbed samples are
typically drawn from Gaussian distributions with a mean of zero and a standard
deviation equal to the prior emission uncertainty in each grid cell. Ensemble runs of the
CMAQ model were subsequently performed to propagate the background errors with
each ensemble sample of state vectors.
In the analysis step, the ensemble mean $\overline{X^a}$ of the analyzed state is regarded as the best
estimate of emissions, which is obtained by updating the background ensemble mean
through the following equations:
$$\overline{X^a} = \overline{X^b} + \mathbf{K}(\mathbf{y} - H\overline{X^b}) \tag{1}$$

$$\mathbf{K} = P^b H^T (H P^b H^T + R)^{-1} \tag{2}$$

where $y$ is the observational vector; $H$ represents the observation operator mapping
model space to observation space; The expression $y - H\overline{X^b}$ quantifies the disparities
between simulated and observed concentrations; $P^b H^T$ illustrates how uncertainties in
emissions relate to uncertainties in simulated concentrations; The Kalman gain matrix
$K$, dependent on background error covariance $P^b$ and observation error covariance $R$,
determines the relative contributions to the updated analysis.
State variables for emissions include $NO_x$ and NMVOCs. To reduce the degree of
freedom in the analysis and avoid the difficulty associated with estimating spatio-
temporal variations in background errors for individual species, we focus on optimizing
the lumped total NMVOC emissions. During the forecast step, we differentiate
individual NMVOC species emissions from the total NMVOC emissions using bottom-
up statistical information. For a consistent comparison between simulations and
observations, model-simulated $NO_2$ were diagnosed at the time and location of surface
$NO_2$ measurements, whereas model-simulated HCHO was horizontally sampled to
align with TROPOMI HCHO VCD retrievals, and subsequently integrated vertically.
In this study, the DA window was set to one day and daily TROPOMI HCHO columns
were utilized as observational constraints in our inversion framework. The ensemble
size was set to 50 to strike a balance between computational cost and inversion accuracy.
To reduce the impact of unrealistic long-distance error correlations, the Gaspari and
Cohn function (Gaspari and Cohn, 1999) was utilized as covariance localization to
ensure the meaningful influence of observations on state variables within a specified



cutoff radius, while mitigating their negative impacts on distant state variables. The
optimal localization scale is interconnected with factors such as the assimilation
window, the dynamic system, and the lifetime of chemical species. Given the average
wind speed of 2.8 m/s (Table S2) and a DA window of 1 day, the localization scales for
$NO_2$ and HCHO, both characterized as highly reactive species with lifespans of just a
few hours, were set to 150 km and 100 km, respectively.

## 2.2 Observation Data and Errors

Considering the availability of HCHO data, we utilized daily offline retrievals of
tropospheric HCHO columns from Sentinel-5P (S5P) L3 TROPOMI data obtained
through Google Earth Engine (De Smedt et al., 2018). The S5P satellite follows a near-
polar sun-synchronous orbit at an altitude of 824 km with a 17-day repeating cycle. It
crosses the Equator at 13:30 local solar time (LST) on the ascending node. The spatial
resolution at nadir was refined to $3.5 \times 5.5\ \text{km}^2$ on 6 August 2019. Following the
recommendations in the S5P HCHO product user manual, we filtered the source data
to exclude pixels with qa_value less than 0.5 for HCHO column number density and
0.8 for aerosol index (AER_AI). The remaining high-quality pixels with minimal
snow/ice or cloud interference are averaged to 27-km grids. Figure 1b illustrates the
coverage and data amount of TROPOMI HCHO retrievals in August 2022 after
processing. Although the distribution of filtered data exhibits spatial non-uniformity,
most grid cells have observational coverage for over half of the time, particularly in the
southern region of China where NMVOC emissions are higher. We assigned
measurement errors of 30% to TROPOMI HCHO columns based on validation against
a global network of 25 ground-based Fourier transform infrared (FTIR) column
measurements (Vigouroux et al., 2020). The representation error can be disregarded
because the model's resolution significantly surpasses that of the TROPOMI pixels.
To address the chemical feedback among VOC-$NO_x$-$O_3$, we also simultaneously
optimized $NO_x$ emissions by assimilating in-situ $NO_2$ observations. The extensively
covered and high-precision monitoring network can provide sufficient constraints for
emission inversion (Figure 1a). Hourly averaged surface $NO_2$ observations from
national control air quality stations obtained from the Ministry of Ecology and
Environment of the People's Republic of China (http://106.37.208.228:8082/, last
access: 5 May 2023). In case where multiple stations are located within the same grid,
a random site is chosen for validation, while the remaining sites are averaged to mitigate



the impact of error correlation (Houtekamer and Zhang, 2016) for assimilation. In total,
1276 stations were chosen for assimilation and an additional 425 independent stations
were selected for verification (Figure 1a). The observation error covariance matrix $\boldsymbol{R}$
incorporates contributions from both measurement and representation errors. The
measurement error is defined as $\varepsilon_0 = 1.0 + 0.005 \times \Pi_0$, where $\Pi_0$ represents the
observed $NO_2$ concentration. Following the approach of Elbern et al. (2007) and Feng
et al. (2018), the representative error is defined as $\varepsilon_r = \gamma \varepsilon_0 \sqrt{\Delta l / L}$, where $\gamma$ is a tunable
parameter (here, $\gamma$=0.5), $\Delta l$ is the grid spacing (27 km), and $L$ is the radius (here, $L$=0.5)
of the observation's influence area. The total observation error ($r$) was defined as $r =$
$\sqrt{\varepsilon_0{}^2 + \varepsilon_r{}^2}$. The observation errors are assumed to be uncorrelated so that $\boldsymbol{R}$ is a
diagonal matrix.



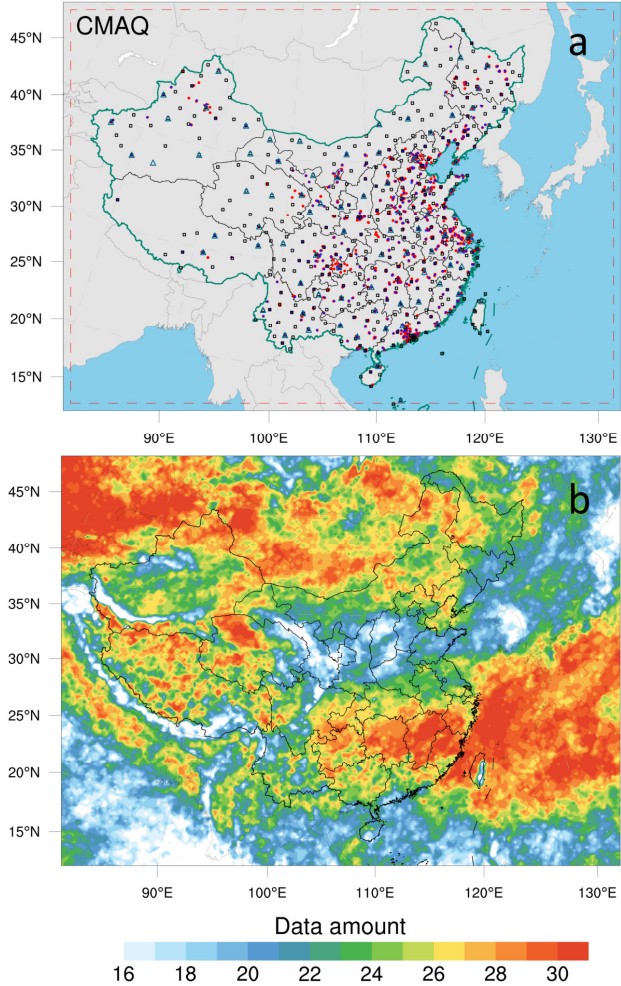


**Figure 1**. Model domain and observation network (a) and data amount of TROPOMI
HCHO retrievals during August 2022 in each grid (b). The red dashed frame delineates
the CMAQ computational domain; black squares denote surface meteorological
measurement sites; navy triangles indicate sounding sites (Text S1), and red and blue
dots represent air pollution measurement sites.

**2.3 Prior Emissions and Uncertainties**

The prior anthropogenic $NO_x$ and NMVOC emissions for China were obtained from
the most recent Multi-resolution Emission Inventory for China of 2020 (MEIC,
http://www.meicmodel.org/, last access: 8 May 2023) (Zhang et al., 2009). For
anthropogenic emissions outside China, we utilized the mosaic Asian anthropogenic
emission inventory (MIX) for the base year of 2010 (Li et al., 2017b). The daily



emission inventory, which was arithmetically averaged from the combined monthly
emission inventory, was employed as the first guess. Ship emissions were derived from
the shipping emission inventory model (SEIM) for 2017, which was calculated based
on the observed vessel automatic identification system (Liu et al., 2017). Biomass
burning emissions were retrieved from the Global Fire Emissions Database version 4.1
(GFEDv4, https://www.globalfiredata.org/, last access: 8 May 2023) (van der Werf et
al., 2017; Mu et al., 2011). Biogenic $NO_x$ and NMVOC emissions were calculated using
the Model of Emissions of Gases and Aerosols from Nature (MEGAN) developed by
Guenther et al. (2012).
As previously mentioned, the optimized emissions are transferred to the next DA
window as prior emissions for iterative inversion. For biogenic emissions, it is
decomposed into hourly scales based on the daily varying temporal profiles in MEGAN
as model inputs. Daily emission variations will largely dominate the uncertainty in
emissions. Taking into account compensating for model errors and avoiding filter
divergence, we consistently applied an uncertainty of 25% to each model grid of $NO_x$
emissions at each DA window, as in Feng et al. (2020). NMVOC emissions typically
exhibit greater uncertainties compared to $NO_x$ emissions (Li et al., 2017b). Based on
model evaluation, the uncertainty of NMVOC emissions was set to 40% (Kaiser et al.,
2018; Souri et al., 2020; Cao et al., 2018). This study also addresses uncertainties in
emissions for CO, $SO_2$, primary $PM_{2.5}$, and coarse $PM_{10}$ to consider the chemical
feedback between different species following Feng et al. (2023).

## 3 Experimental Design

Before implementing the emission inversion, a relatively perfect initial field is
generated at 0000 UTC on August 01 through conducting a 5-day simulation with 6-
hour interval 3D-Var data assimilation. Subsequently, daily emissions are continuously
updated over the entire month of August (EMDA). To validate the posterior emissions
of $NO_x$ and NMVOCs, we compared two parallel forward simulation experiment with
$NO_2$ and HCHO measurements, denoted as CEP and VEP, corresponding to prior and
posterior emission scenarios, respectively. To investigate the impact of optimizing
NMVOC emissions on the secondary production and loss of surface $O_3$, a forward
simulation experiment (CEP1) was conducted with the prior NMVOC emissions and
the posterior $NO_x$ emissions. Additionally, we designed three sensitivity experiments to
investigate the robustness of the constrained NMVOC emissions. EMS1 involved



doubling the background error from 40% to 80% to investigate the influence of
background error settings. EMS2 aimed to evaluate the effect of observational data
retrieval errors on emission estimates, in which HCHO columns were empirically bias-
corrected based on error characteristics (Souri et al., 2021). EMS3 aimed to illustrate
the significance of optimizing $NO_x$ emission in quantifying $VOC-O_3$ chemical reactions.
In this experiment, $NO_x$ emissions were not optimized. Two forward modelling
experiments (CEP2 and CEP3) were also performed using the posterior emissions of
EMS2 and EMS3 to evaluate their performance. All experiments employ identical
meteorological fields, as well as the same gas-phase and aerosol modules. Table 1
summarizes the different emission inversion and validation experiments conducted in
this study.
**Table 1**. The assimilation, sensitivity, and validation experiments conducted in this
study.

| Exp.Type | Exp. Name | NMVOC emissions | $NO_x$ emissions | Assimilated HCHO retrievals |
|---|---|---|---|---|
| Assimilation | EMDA | MEIC 2020 and MEGAN for August (the first DA window), optimized emissions of the previous window (other DA windows) | MEIC 2020 and MEGAN for August (the first DA window), optimized emissions of the previous window (other DA windows) | Default |
| Sensitivity | EMS1 | Same as EMDA but with doubled default uncertainty | Same as EMDA | Default |
| | EMS2 | Same as EMDA | Same as EMDA | Reduce by 25% in regions with observations $< 2.5 \times 10^{15}$ molec $cm^{-2}$ and increase by 30% in regions with observations $> 8 \times 10^{15}$ molec $cm^{-2}$ |
| | EMS3 | Same as EMDA | MEIC 2020 and MEGAN for August | Default |
| Validation | CEP | MEIC 2020 and MEGAN for August | MEIC 2020 and MEGAN for August | \ |
| | VEP | Posterior emissions of EMDA | Posterior emissions of EMDA | \ |
| | CEP1 | Same as CEP | Posterior emissions of EMDA | \ |
| | CEP2 | Posterior emissions of EMS2 | Posterior emissions of EMS2 | \ |
| | CEP3 | Posterior emissions of EMS3 | Same as CEP | \ |



## 4 Results

### 4.1 Inverted Emissions

Figure 2 shows the spatial distribution of temporally averaged prior and posterior emissions, along with their differences, in NMVOC emissions. Hotspots of prior NMVOC emissions were prevalent across much of central and southern China. However, posterior NMVOC emissions were predominantly concentrated in the NCP, Yangtze River Delta (YRD), PRD, and Sichuan Basin (SCB), characterized by high levels of anthropogenic activity. High emissions are also located in parts of central and southern China with warm climate favorable for emitting biogenic NMVOCs. Employing TROPOMI HCHO observations as constraints led to widespread decreases of approximately 60–70% over these areas, indicating a large substantial of biogenic NMVOC emissions. In northwestern China, there was a moderate increase in NMVOC emissions. Validation efforts against 28 NDACC FTIR stations reported that TROPOMI generally displays a negative bias of -30% for HCHO concentrations exceeding $8\times10^{15}$ molec cm$^{-2}$, while a positive bias of 34% is observed at clean sites with HCHO concentrations below $2.5\times10^{15}$ molec cm$^{-2}$ (Lambert et al., 2023). Comparisons with MAX-DOAS measurements yielded similar biases. A potential significant bias in polluted regions could exacerbate the emission reduction. Nevertheless, the large magnitude of emission reductions of 50.2% in our inversion is comparable to studies in southern China (Bauwens et al., 2016; Zhou et al., 2023), southeastern US (Kaiser et al., 2018), Africa (Marais et al., 2014), India (Chaliyakunnel et al., 2019), Amazonia (Bauwens et al., 2016), and parts of Europe (Curci et al., 2010), but opposite to the large-scale emission increase over China in Cao et al. (2018). For NO$_x$ (Figure S1), the nationwide total emissions decreased by 10.2%, with the main reductions concentrated in the NCP, YRD, parts of Central China, and most key urban areas.





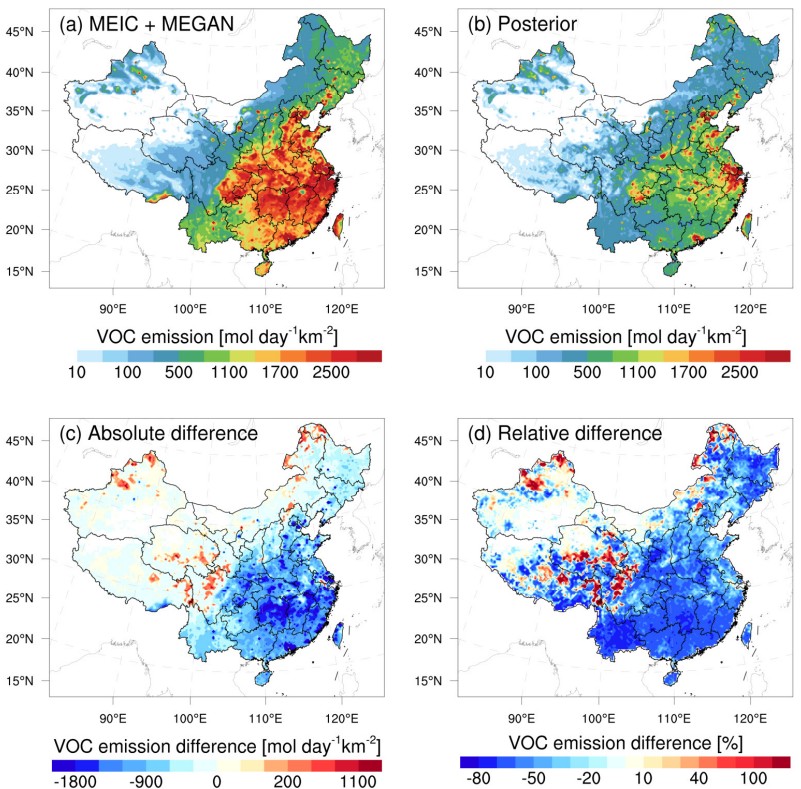

**Figure 2**. Spatial distribution of the time-averaged (a) prior emissions (MEIC 2020 + EMGAN), (b) posterior emissions, (c) absolute difference (posterior minus prior), and (d) relative difference of NMVOCs over China.

Table 2 shows the changes in emissions of biogenic NMVOCs across different land cover types (Figure S2) after inversion. The most significant reduction in biogenic emissions occurred within woody savannas, accounting for 26.9% of the overall reduction, followed by savannas and croplands, accounting for 21.2% and 17.2% respectively. Among all vegetation types, the broadleaf evergreen forests, recognized as the primary source of isoprene emission (Wang et al., 2021a), presented the greatest uncertainty, with NMVOC emissions experiencing a significant reduction of 66.2%. Standard emission rates in MEGAN are derived from leaf- or canopy-scale flux measurements and extrapolated globally across regions sharing similar landcover characteristics, based on very limited observations (Guenther et al., 1995). This methodology introduces biases due to the large variability in emission rates among plant species. Furthermore, DiMaria et al. (2023) optimized the temperature response

  



within MEGAN with ground-based constraints, increasing the model's temperature
sensitivity by a factor of five over the Amazonian. Opacka et al. (2022) optimized the
empirical parameter in the MEGANv2.1 soil moisture stress algorithm, resulting in
significant reductions in isoprene emissions and providing better agreement between
modelled and observed HCHO temporal variability in the central U.S. These findings
demonstrate that uncertainties in MEGAN parameterization also have significant
implications for NMVOC emission modeling.
**Table 2**. Prior and posterior biogenic NMVOC emissions, as well as their differences
for different land cover types.

| Land cover type | Prior | Posterior | Difference |
|---|---|---|---|
| | Mmol/month | Mmol/month | Mmol/month (%) |
| Evergreen needleleaf forests | 955.7 | 549.3 | -406.4 (-42.5) |
| Evergreen broadleaf forests | 13985.1 | 4728.2 | -9256.8 (-66.2) |
| Deciduous needleleaf forests | 46.6 | 48.8 | 2.2 (4.7) |
| Deciduous broadleaf forests | 8335.5 | 3487.4 | -4848.1 (-58.2) |
| Mixed forests | 8731.0 | 3961.7 | -4769.4 (-54.6) |
| Closed shrublands | 9.7 | 3.7 | -6.0 (-61.5) |
| Open shrublands | 21.3 | 8.6 | -12.8 (-59.8) |
| Woody savannas | 39327.2 | 16925.2 | -22402.0 (-57.0) |
| Savannas | 28319.7 | 10629.4 | -17690.3 (-62.5) |
| Grasslands | 16912.7 | 14269.6 | -2643.1 (-15.6) |
| Permanent wetlands | 286.1 | 115.4 | -170.8 (-59.7) |
| Croplands | 25537.8 | 11215.5 | -14322.2 (-56.1) |
| Cropland-natural vegetation mosaics | 10894.7 | 4289.8 | -6605.0 (-60.6) |
| Sparsely vegetated | 1814.7 | 1644.0 | -170.6 (-9.4) |

**4.2 Evaluations for Posterior Emissions**
The NO$_x$ emissions were first evaluated by indirectly comparing the forward simulated
NO$_2$ concentrations with measurements. As shown in Figure S3, the CEP with prior
emissions exhibited positive biases in eastern China and negative biases in western
China. However, when posterior emissions were used in the VEP, a substantial



improvement in simulation performance was observed. Biases were limited to within
±3 µg m⁻³, and correlation coefficients exceeded 0.7 across the entire region. Figure 3
presents the simulated HCHO VCDs using prior and posterior NMVOCs emissions,
along with their associated biases. Both experiments showed high VCDs over central
and eastern China, especially in the YRD and SCB. However, the CEP displayed
substantial overestimation across most of mainland China, with the largest bias
reaching 12 × 10¹⁵ molec cm⁻² in Central China. Conversely, the VEP demonstrated
notable improvements in both the magnitude and spatial distribution of simulated
HCHO columns after the inversion compared to TROPOMI retrievals. More than 84%
of the areas exhibited biases of less than 1 × 10¹⁵ molec cm⁻², and no significant spatial
variation was observed. Overall, the biases in simulated HCHO VCDs decreased by
75.7% after the inversion. These results emphasize the efficiency of our system in
reducing uncertainty in both NO$_x$ and NMVOC emissions.

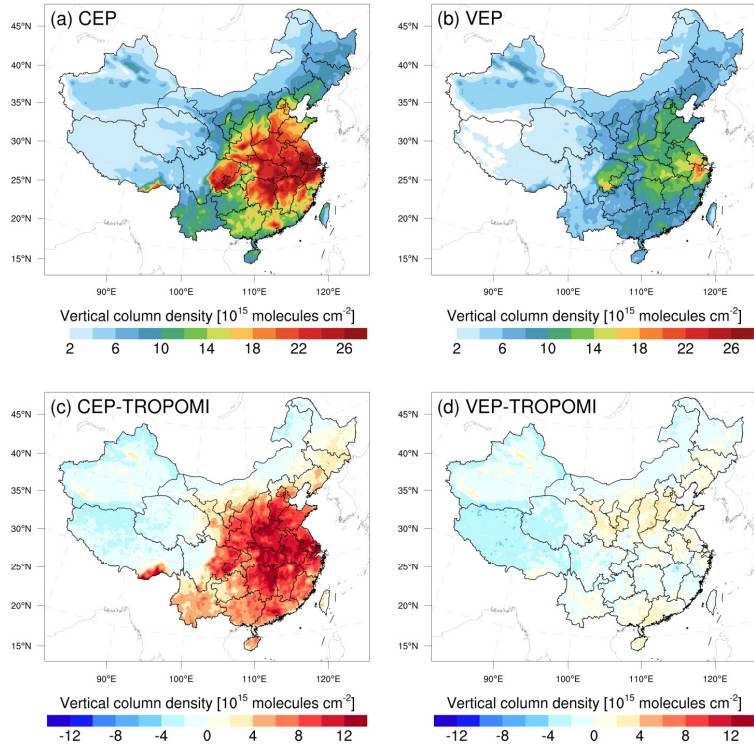


**Figure 3**. Simulated HCHO vertical column densities using prior (a) and posterior (b)
NMVOC emissions, along with their biases (c and d) against TROPOMI measurement.
All model results were sampled at TROPOMI overpass time.



### 4.3 Implications for Surface O₃

Figure 4 shows the spatial distribution of the mean bias (BIAS), root mean square error (RMSE), and correlation coefficient (CORR) for simulated $O_3$ concentrations in the CEP1 and VEP experiments compared to assimilated observations. Beyond the northwestern region of China, the CEP1 exhibited significant overestimation throughout the entire area, with a BIAS of 20.5 μg m$^{-3}$. By intercomparing 14 state-of-the-art CTMs with $O_3$ observations within the framework of the MICS-Asia III, Li et al. (2019) identified a substantial overestimation of annual surface $O_3$ in East Asia, ranging from 20 to 60 μg m$^{-3}$. Notably, the NCP exhibited substantial overestimations, with most models overestimating $O_3$ by 100–200% during May–October. In the VEP, the modeled $O_3$ chemical production were alleviated, especially in the southern regions of China where NMVOC emissions had significantly decreased. Overall, observation-constrained NMVOC emissions resulted in a 49.3% decrease in the BIAS, bringing it down to 10.4 μg m$^{-3}$. Additionally, the RMSE showed noticeable improvement due to the assimilation of HCHO observation, reducing the value from 30.9 to 23.3 μg m$^{-3}$. Despite a significant reduction in NMVOC emissions after inversion, notable overestimations persisted in northern provinces such as Liaoning, Hebei, Shanxi, and Shaanxi. This may be attributed to limited NMVOC constraints resulting from insufficient observations during the study period (Figures 1b and 3d). The remaining discrepancies between simulations and observations can be attributed to the combined results of intricate urban-rural sensitivity regimes and $O_3$ photochemistry reactions, which may not be comprehensively represented by CMAQ model, masking any potential improvement expected from the constrained emissions. The CORR was comparable between the CEP1 and VEP experiments, reflecting that the CMAQ model effectively simulated the temporal variation of $O_3$ concentrations. The biases at the independent sites were similar to those at the assimilated sites (Figure S4). In comparison to CEP1, the decreasing ratios in BIAS and RMSE in VEP were 46.7% and 23.4%, respectively.



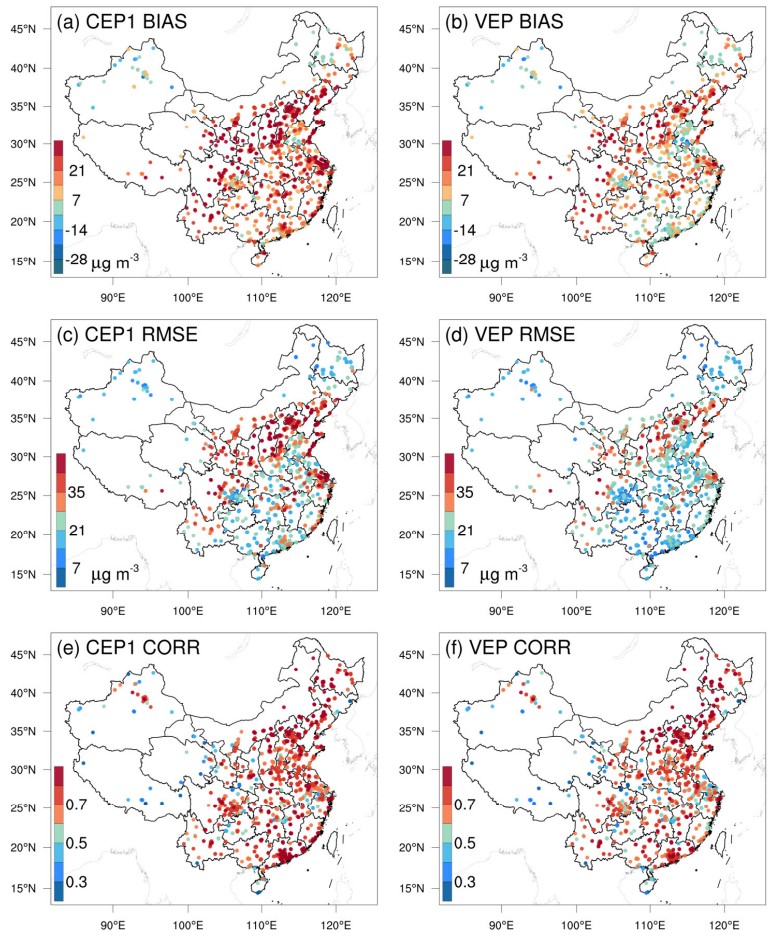

425

**Figure 4**. Spatial distribution of mean bias (BIAS, a and b), root mean square error (RMSE, c and d), and correlation coefficient (CORR, e and f) for simulated O$_3$ using prior (left, CEP1) and posterior (right, VEP) emissions, respectively, against assimilated observations.

Figure 5 shows the time series of simulated and observed hourly O$_3$ concentrations and their RMSEs, verified against surface monitoring sites. The VEP achieved better representations of diurnal O$_3$ variations compared with those in the CEP1, especially excelling in reproducing elevated O$_3$ concentrations at noon. Constraining the NMVOC emissions also led to better model simulations in terms of RMSE throughout the entire study period. Overall, the assimilation of HCHO column observations effectively reduced NMVOC emission uncertainties and consequently improved simulations of



HCHO and $O_3$. These improvements hold promise for further research into the
implications of emission optimizations on regional $O_3$ photochemistry.

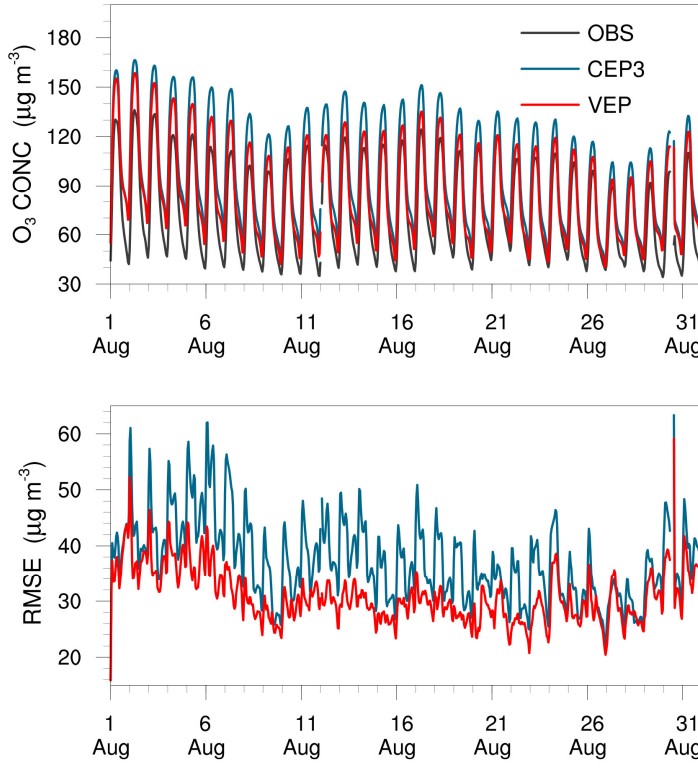


**Figure 5**. Time series comparison of hourly surface $O_3$ concentrations ($\mu g\ m^{-3}$) and

RMSE ($\mu g\ m^{-3}$) from CEP1 and VEP experiments against all observations.
As crucial $O_3$ precursors, the abundance of NMVOCs plays a significant role in
modulating $O_3$ production. Here we employed the IRRs to elucidate changes related to
$O_3$ production and loss at the surface, stemming from constrained $NO_x$ and NMVOC
emissions. Figure 6 illustrates comparisons of the simulated maximum daily 8-hour
average (MDA8) surface $O_3$ levels and net reaction rates before and after the inversion.
The CEP1 exhibited an overestimation of $O_3$ levels, with a BIAS of 22.6% compared
to observed $O_3$ concentrations. This overestimation corresponded to the high net
chemical rates of $O_3$ in these areas (Figure S5). After inversion, $O_3$ net rates mitigated
in most regions. Consequently, the VEP experiment yielded results that closely aligned
with observations, with a BIAS of 9.2%. Referring to Figure 6e and 6f, differences in
production rates of $O_3$ closely track the changes in the NMVOC emissions (Figure 2).



The discrepancies in specific regions may be attributed to the complex nonlinear
relationships associated with $O_3$ and its precursors, which depend on prevailing
chemical regimes and regional transport. Additionally, changes in $O_3$ production
predominantly drive the overall decrease in $O_3$ concentrations, outweighing changes in
$O_3$ loss.

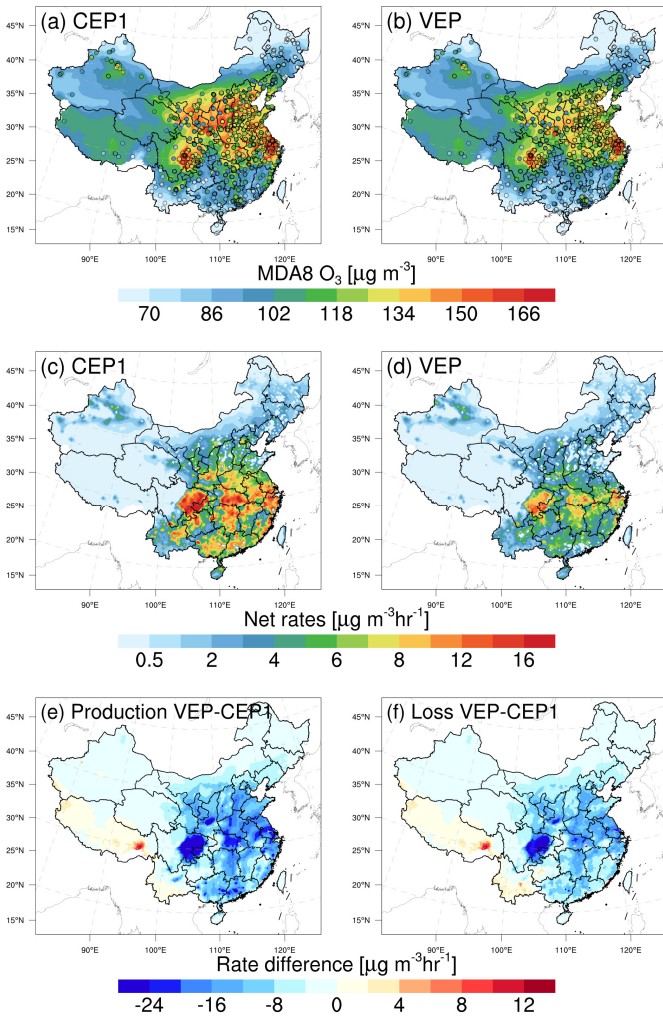


**Figure 6**. Comparisons of (a, b) simulated maximum daily 8-hour average (MDA8) $O_3$
concentrations, (c, d) net reaction rates, (e, f) and differences in production and loss
rates between CEP1 and VEP experiments at the surface. Surface MDA8 $O_3$ values
(circles) from the national control air quality stations were overlaid



Figure 7 shows the differences in the six principal pathways responsible for $O_3$ loss and
formation, when comparing simulations employing prior and posterior emissions. The
reactions of $HO_2 + NO$ and $RO_2 + NO$ are treated as the pathways leading to $O_3$
formation, whereas $O_3$ loss involves reactions including $NO2 + OH$, $O_3 + HO_2$, $O_3 +$
$NMVOCs$, and $O1D + H_2O$ (Wang et al., 2019). Our analysis was focused on the time
frame from 12:00 to 18:00 according to China standard time (CST). The differences
were computed by subtracting the simulation with posterior emissions from those with
prior emissions. Following the emission of NMVOCs, they undergo rapid oxidation by
atmospheric hydroxyl (OH) radicals. Due to the substantial decrease in NMVOC
emissions, there was a reduction in the production of hydroperoxy radicals ($HO_2$) and
organic peroxy radicals ($RO_2$) (Figure S6). Consequently, this reduction in $HO_2/RO_2$
levels, coupled with their reaction with NO, resulted in diminished $O_3$ production
(Figures 7a and 7b). A strong correlation was observed between changes in $O_3$
production via the $RO_2 + NO$ reaction and NMVOC emissions (Figure 2), consistent
with the findings of Souri et al. (2020). Typically, in NMVOC-rich environments, a
decrease in NMVOC emissions boosts OH concentrations. Consequently, we noted an
enhancement in the $NO_2 + OH$ reaction in the eastern and central regions of China. In
response to heightened $HO_x$ concentrations over these areas, an increased $O_3$ loss
through the $O_3 + HO_x$ pathway was observed. Furthermore, we detected a substantial
decrease in $O_3$ loss through reactions with NMVOCs, especially in the southern China,
where substantial isoprene emissions are prevalent. This reduction was primarily
attributable to the decrease in NMVOC and $O_3$ levels. While the $NMVOC + O_3$ reaction
proceeds at a substantially slower rate $NMVOC + OH$, this specific chemical pathway
remains significant in oxidizing NMVOC and forming $HO_x$ in forests areas (Paulson
and Orlando, 1996). The difference in $O1D + H_2O$ is primarily driven by the decrease
of $O_3$ photolysis. Although the rate of $O_3$ loss decreases in some chemical pathways,
overall, the rate of $O_3$ production dominates the changes in $O_3$ concentration.



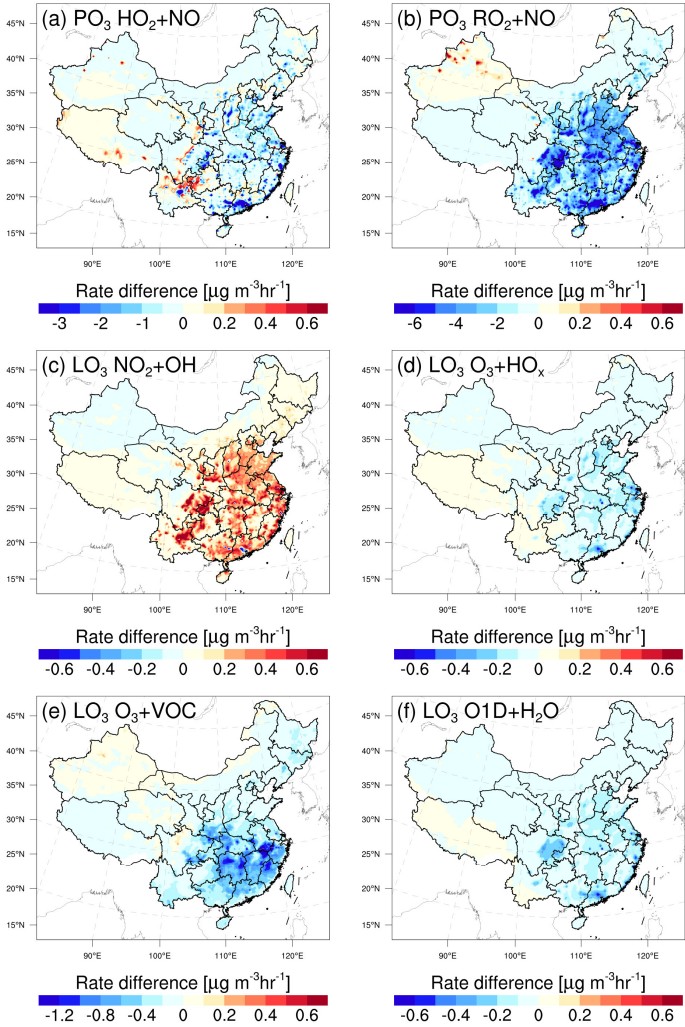

**Figure 7**. Differences in six major pathways of $O_3$ production and loss between CEP1 and VEP experiments at the surface. Time period: August 2022, 12:00–18:00 CST. $PO_3$ and $LO_3$ represent the pathways of $O_3$ formation and loss, respectively.

## 4.4 Discussions

The prior NMVOC emissions were found to be overestimated relative to the top-down constraints from TROPOMI HCHO retrievals. The results of the top-down inversion may be susceptible to uncertainties related to the inversion configuration and observational data. Particularly, background error settings affect the relative weighting of prior and observation to posterior emissions, which may potentially introduce



considerable uncertainty into the spatial patterns and magnitudes of the NMVOC
emission inversion. Another critical concern pertains to HCHO retrieval errors.
Correcting the low TROPOMI HCHO column biases would result in elevated posterior
emissions, while the opposite holds true. To investigate the impact of background error
on emission inversion, a sensitivity test (EMS1) was conducted, doubling the
background error to 80%. Compared with the base inversion, the sensitivity test
produced a noticeable increase in posterior NMVOC emissions in southwestern China,
especially in Tibet. In contrast, emissions in eastern China exhibited a slight decrease
(Figure S7). This can be expected, as the inversion is more inclined to deviate from the
a priori due to decreased confidence. However, at a national scale, the difference
between the two posterior emissions was nearly negligible. The substantial disparities
over the Tibetan Plateau between the two inversions can be attributed to the horizontal
HCHO inhomogeneity caused by mountain terrain and the relatively low signal-to-
noise ratio in the TROPOMI satellite data in the background atmosphere (Cheng et al.,
2023), resulting in the inclusion of more outliers in the inversion (Su et al., 2020).
Nevertheless, the discrepancies in NMVOC emission estimates amounted to a mere
0.2%, suggesting that the posterior emission estimates were not largely affected by the
background error setting. This can be primarily attributed to the superiority of the 'two-
step' inversion strategy employed within the RAPAS system.
Due to the spatiotemporal variability in retrieval errors, directly incorporating
observations into an inversion system remains a challenging task. Based on the biases
outlined in Vigouroux et al. (2020), another sensitivity test (EMS2) addressed the
existing biases in TROPOMI HCHO by reducing measurements by 25% ($<2.5\times10^{15}$
molec cm$^{-2}$) in clean regions and increasing them by 30% ($>=8\times10^{15}$ molec cm$^{-2}$) in
polluted regions. Figure 8 shows that bias-corrected HCHO columns resulted in a slight
decrease in NMVOC emissions in the low-pollution regions of western China, whereas
emissions increased in the high-pollution regions of eastern and central China,
particularly in the SCB and the vicinity of the YRD. In comparison to the EMDA
experiment, the posterior emissions from EMS1 increased by 12.8% (decreased by 43.9%
compared to prior emissions), indicating that the existing retrieval error in HCHO
measurements likely exerts an influence on the estimation of NMVOC emissions,
especially in heavily polluted regions. The results highlight the significance of a
thorough data validation for the HCHO column product. However, the emissions



increase in the EMS2 experiment has slightly deteriorated the performance of O$_3$
simulations in the CEP2.

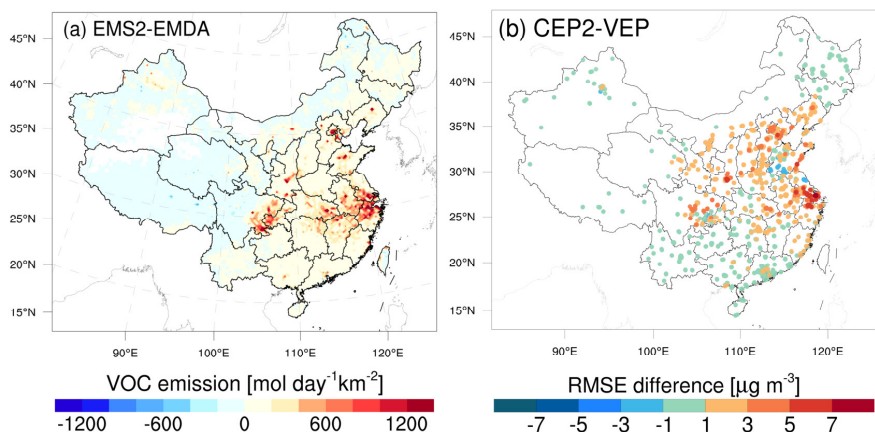


**Figure 8**. Spatial distribution of (a) differences in posterior emissions between EMS2
and EMDA, and differences in (b) RMSE between CEP2 and VEP experiments.
Compared with EMDA, EMS2 reduced the TROPOMI HCHO measurements by 25%
($< 2.5 \times 10^{15}$ molec cm$^{-2}$) in clean regions and increased them by 30% ($> 8 \times 10^{15}$ molec
cm$^{-2}$) in polluted regions.
O$_3$ concentration and NO$_x$ (VOC) emissions are positively correlated in the NO$_x$
(VOC)-limited region and negatively correlated in the VOC (NO$_x$)-limited region (Tang
et al., 2011). Therefore, the uncertainty in NO$_x$ emissions can affect the model's
diagnosis of O$_3$-NO$_x$-VOC sensitivity, thereby introducing substantial model errors in
the HCHO yield from VOC oxidation. In the base inversion experiment (EMDA), we
simultaneously assimilated NO$_2$ and HCHO observations to optimize NO$_x$ and
NMVOC emissions. To evaluate the impact of optimized NO$_x$ emissions on O$_3$-VOC
chemistry, EMS3 disregarded the uncertainty of NO$_x$ and focused solely on optimizing
NMVOC emissions. Compared to the EMDA, in areas where NO$_x$ is significantly
overestimated, NMVOC emissions in the EMS3 have correspondingly decreased
(Figure 9b). This might be due to under high-NO$_x$ conditions, HCHO production occurs
promptly, thereby compensating for the substantial amount of HCHO already present
in the atmosphere by reducing emissions (Chan Miller et al., 2017). Figure S8 shows
comparisons of concentrations and RMSE between the simulations using posterior
emissions from EMS3 and EMDA experiments. Compared to VEP, CEP3 showed a



larger RMSE, highlighting the necessity for simultaneous optimization of NO$_x$ emissions when evaluating the impact of NMVOC emission optimization on O$_3$. Additionally, CEP2 using prior NO$_x$ emissions exhibited lower O$_3$ levels over parts of NCP and YRD, as well as some urban areas (Figure 9c), but with larger biases and RMSEs (Figure 9d). The reduction in NMVOC emissions contributed to a partial decrease in O$_3$ concentration. More significantly, these areas typically align with VOC-limited mechanisms (Wang et al., 2019; Wang et al., 2021b). Therefore, the overestimation of NO$_x$ emissions (Figure S1) excessively inhibits O$_3$ accumulation due to the titration effect, thereby disrupting the evaluation of NMVOC contributions to O$_3$. This substantial disparity also seriously affects O$_3$ source apportionment, precursor-sensitive area delineation, and emissions reduction policy formulation.

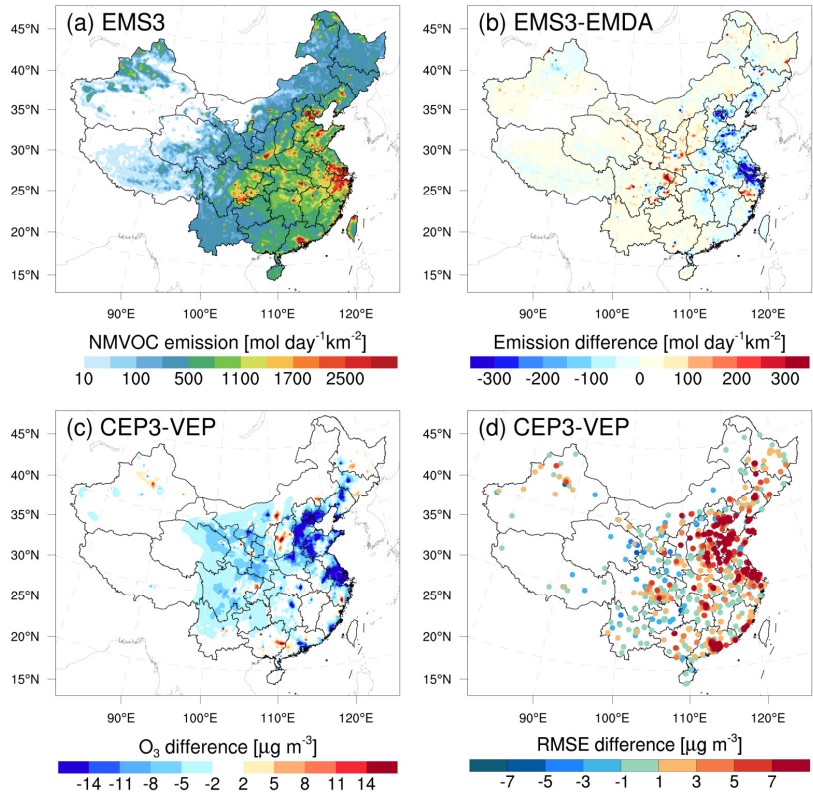

**Figure 9**. Spatial distribution of (a) posterior emissions in the EMS3 experiment, (b) differences in posterior emissions between EMS3 and EMDA, and differences in simulated (c) O$_3$ concentrations and (d) RMSE between CEP3 and VEP experiments. EMS3 did not optimize NO$_x$ emissions compared to EMDA.



## 5 Summary and Conclusions

In this study, we extended the RAPAS assimilation system with the EnKF assimilation algorithm to optimize NMVOC emissions using the TROPOMI HCHO retrievals. Taking the MEIC 2020 for anthropogenic emissions and MEGANv2.1 output for biogenic sources as a priori, NMVOC emissions over China in August 2022 were inferred. Importantly, we implicitly took the chemical feedback among VOC-$NO_x$-$O_3$ into account by simultaneously adjusting $NO_x$ emissions using nationwide in-situ $NO_2$ observations. Furthermore, we quantified the impact of NMVOC emission inversion on surface $O_3$ pollution using the CMAQ-IRR model.

The application of TROPOMI HCHO observations as constraints led to a substantial reduction of 50.2% compared to the prior emissions for NMVOCs. A domain-wide significant decrease was found over central and southern China with abundant forests, especially for the broadleaf evergreen forests, implying a considerable overestimation of biogenic NMVOC emissions. Observation-constrained emissions significantly improved the performance of surface $NO_2$ and HCHO column simulations, reducing biases by 97.4% and 75.7%, respectively. This highlights the effectiveness of the RAPAS in reducing uncertainty in $NO_x$ and NMVOC emissions. Isolating the impact of $NO_x$ emission changes, the posterior NMVOC emissions significantly mitigated the overestimation in prior $O_3$ simulations, resulting in a 49.3% decrease in surface $O_3$ biases. This is mainly attributed to a substantial decrease in the RO2 + NO reaction rate (a major pathway for $O_3$ production) and an increase $NO_2$ + OH reaction rate (a major pathway for $O_3$ loss) during the afternoon, resulting in a decrease in the simulated MDA8 surface $O_3$ concentrations by approximately 15 μg m$^{-3}$.

Sensitivity inversions demonstrate the robustness of top-down emissions to variations in background error settings, yet they are sensitive to HCHO column biases, highlighting the importance of comprehensive validation studies utilizing available remote-sensing data and, if possible, airborne validation campaigns. Moreover, we found that, in comparison to optimizing NMVOC emissions alone, the joint optimization of NMVOC and $NO_x$ emissions can significantly improve the overall performance of $O_3$ simulations. Ignoring errors in $NO_x$ emissions introduces uncertainty in quantifying the impact of NMVOC emissions on surface $O_3$, especially in areas where overestimated $NO_x$ emissions can unrealistically amplify titration effects,



highlighting the necessity of simultaneous optimization of NO$_x$ emissions.


## Data availability

The observations used for assimilation and the optimized emissions in this study can be
accessed at https://doi.org/10.5281/zenodo.10079006 (Feng and Jiang, 2023).

## Author contribution

SF and FJ conceived and designed the research. SF developed the data assimilation
code, analyzed data, and prepared the paper with contributions from all co-authors. FJ
supervised and assisted in conceptualization and writing. TQ, NW, MJ, SZ, JC, FY, and
WJ reviewed and commented on the paper.

## Competing interests

The authors declare that they have no conflict of interest.

## Acknowledgements

This work is supported by the National Key R&D Program of China (Grant No.
2022YFB3904801), the National Natural Science Foundation of China (Grant No:
42305116), the Natural Science Foundation of Jiangsu Province of China (Grant No:
BK20230801), and the Hangzhou Agricultural and Social Development Scientific
Research Project (Grant No: 202203B29). The authors also gratefully acknowledge the
High-Performance Computing Center (HPCC) of Nanjing University for doing the
numerical calculations in this paper on its blade cluster system.

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
