# Peer review of "Constraint of non-methane volatile organic compound emissions with"

_EGUsphere, 2023_

## Referee Comment (RC1)

**Comments to "Constraint of non-methane volatile organic compound emissions with TROPOMI HCHO observations and its impact on summertime surface ozone simulation over China" by Feng et al.**

As the key precursors of Ozone ($O_3$), Non-methane volatile organic compounds (NMVOC)  have an important influence on the formation of photochemical, secondary organic aerosols and organic acids, harming human health. It is important and challenge to accurate estimate the spatiotemporal distribution of NMVOC emissions. This study presents the NMVOC emissions over China based on EnKF method by assimilating TROPOMI HCHO retrievals. Authors also optimize NOx emissions to reduce the influence of VOC-NOx-$O_3$ chemical feedback. The results showed that the forecast experiment with posterior NMVOC emissions reduced the uncertainty of HCHO and concentrations simulation. And the impact on surface $O_3$ simulation with prior and posterior NMVOC emissions was analyzed. The results will help to improve model forecasts of HCHO, NOx, and $O_3$ concentrations and contribute to design suitable emission reduction policies.

However, the structure of the article should be revised. Authors conducted four set of DA experiments and five set of forecast experiments. They discuss the influence of background error and observation error on the effect of optimizing HCHO emissions. And They also analyzed the impact on surface $O_3$ simulation with prior and posterior NMVOC emissions. Thus, there are too many goals in the study, and it is difficult for readers to remember the setting of these nine experiments. I suggested to delete the discussion about the influence of background error (**B**) and observation error (**R**) on the effect of optimizing HCHO emissions in the section 4.4. It would be nice to discuss the influence of the **B** and **R** when introducing the EnKF method and explain why authors design the **B** and **R** to optimize NMVOC emissions in this study.

There are several issues that need to be addressed.

**specific comments:**

1. Line 40: It should be "Compared with the forecast experiment with prior emission, the forecast with posterior ...". The statement should be revised.

2. Line 42: "Moreover" should be deleted. And the statement also should be revised

3. Line 176: What did you consider about the boundary condition of NMVOC and NOx?

4. Line 204~207: Did author consider about the correction of NOx and NMVOCs in the DA system?

5. Line 209~210: As $NO_2$ is a kind of short lifetime gas, the concentration of surface $NO_2$ measurements not only present $NO_2$, but also may include NOx. What did you consider about the influence of $NO_2$ observation uncertain on optimizing NOx emissions?

6. Line 265: It would be better to use mosaic diagram to present the data amount of TROPOMI HCHO.

7. Line 299: Please added the year of the study period.

8. Line 307~314: The background error covariance is implicitly expressed in the EnKF method. How did author implement EMS1 experiment in the DA system? And it would be better to introduce EMS1-3 experiment follow the EMDA, making the text description consistent with the Table1.

9. Line 324 and 351: "prior and posterior emissions" should be "prior and posterior NMVOC emissions", and "EMGAN" should be "MEGAN".

10. Line 440-441, Figure 5: It is difficult for readers to remember the setting of experiments. And I think that "CEP3" should be "CEP1" in the Fig. 5a?

11. Line 515-518: The background errors and observation errors play an important role in the DA system. It would be better to give a detailed explanation of why the difference in two posterior NMVOC emissions was small by using 'two-step' inversion strategy in the DA system.

---

## Author Comment (AC1)

**Responses to the comments of Reviewer #2:**

We would like to thank the anonymous referee for his/her comprehensive review and valuable suggestions. These suggestions help us to present our results more clearly. In response, we have made changes according to the referee's suggestions and replied to all comments point by point. All the page and line number for corrections are referred to the revised manuscript, while the page and line number from original reviews are kept intact.

**General comments:**

As the key precursors of Ozone ($O_3$), Non-methane volatile organic compounds (NMVOC) have an important influence on the formation of photochemical, secondary organic aerosols and organic acids, harming human health. It is important and challenge to accurate estimate the spatiotemporal distribution of NMVOC emissions. This study presents the NMVOC emissions over China based on EnKF method by assimilating TROPOMI HCHO retrievals. Authors also optimize NOx emissions to reduce the influence of VOC-NOx-O3 chemical feedback. The results showed that the forecast experiment with posterior NMVOC emissions reduced the uncertainty of HCHO and concentrations simulation. And the impact on surface O3 simulation with prior and posterior NMVOC emissions was analyzed. The results will help to improve model forecasts of HCHO, NOx, and O3 concentrations and contribute to design suitable emission reduction policies.

However, the structure of the article should be revised. Authors conducted four set of DA experiments and five set of forecast experiments. They discuss the influence of background error and observation error on the effect of optimizing HCHO emissions. And They also analyzed the impact on surface O3 simulation with prior and posterior NMVOC emissions. Thus, there are too many goals in the study, and it is difficult for readers to remember the setting of these nine experiments. I suggested to delete the discussion about the influence of background error (B) and observation error (R) on the effect of optimizing HCHO emissions in the section 4.4. It would be nice to discuss the

influence of the B and R when introducing the EnKF method and explain why authors design the B and R to optimize NMVOC emissions in this study.

**Response:** We appreciate the reviewer for his/her constructive and up-to-point comments. We have further elaborated on the rationale behind the selection of observational and background error settings. We also briefly discussed the influence of TROPOMI retrieval errors and background errors on optimizing HCHO emissions in Section 2.2 and 2.3, respectively. We also deleted the aforementioned discussion in Section 4.4. Correspondingly, we removed the EMS1, EMS2, and CEP2 experiments from the original manuscript, and renamed the EMS3 experiment to EMS, and renamed the CEP3 experiment to CEP2 in the revised manuscript.
See lines 249-257, pages 8-9.

"Based on validation against a global network of 25 ground-based Fourier transform infrared (FTIR) column measurements (Vigouroux et al., 2020), TROPOMI HCHO overestimates by 25% ($<2.5\times10^{15}$ molec cm$^{-2}$) in clean regions and underestimates by 30% ($>=8\times10^{15}$ molec cm$^{-2}$) in polluted regions. Therefore, we set the measurement error to 30%. To evaluate the effect of observational data retrieval errors on emission estimates, we conducted a sensitivity experiment in which HCHO columns were empirically bias-corrected according to the error characteristics described above (Figure S1). The posterior emissions increased by 12.8% compared to those in the base experiment (EMDA), indicating that the existing retrieval error in HCHO measurements likely exerts an influence on the estimation of NMVOC emissions."
See lines 312-317, page 12.

"… …Based on model evaluation, the uncertainty of NMVOC emissions was set to 40% (Kaiser et al., 2018; Souri et al., 2020; Cao et al., 2018). A sensitivity experiment involving a doubling of the prior uncertainty (80%) revealed that the differences in posterior NMVOC emissions amounted to a mere 0.2% (Figure S2). The implementation of a 'two-step' inversion strategy allows for the timely correction of residual errors from the previous assimilation window in the current window, thus ensuring that the RAPAS system has a relatively low dependence on prior uncertainty

settings. This study also addresses uncertainties… …"

"… …Additionally, we designed a sensitivity experiment (EMS) to illustrate the significance of optimizing $NO_x$ emissions in quantifying VOC-$O_3$ chemical reactions. In this experiment, $NO_x$ emissions were not optimized. To validate the posterior emissions of $NO_x$ and NMVOCs in EMDA, we compared two parallel forward simulation experiments, denoted as CEP and VEP, corresponding to prior and posterior emission scenarios, respectively, against $NO_2$ and HCHO measurements. To investigate the impact of optimizing NMVOC emissions on the secondary production and loss of surface $O_3$, a forward simulation experiment (CEP1) was conducted with the prior NMVOC emissions and the posterior $NO_x$ emissions. Another forward modelling experiment (CEP2) used the posterior emissions of EMS to evaluate its performance.… …"

Table 1. The assimilation, sensitivity, and validation experiments conducted in this study.

| Exp.Type | Exp. Name | NMVOC emissions | $NO_x$ emissions |
|---|---|---|---|
| Assimilation | EMDA | MEIC 2020 and MEGAN for August (the first DA window), optimized emissions of the previous window (other DA windows) | MEIC 2020 and MEGAN for August (the first DA window), optimized emissions of the previous window (other DA windows) |
| Sensitivity | EMS | Same as EMDA | MEIC 2020 and MEGAN for August |
| Validation | CEP | MEIC 2020 and MEGAN for August | MEIC 2020 and MEGAN for August |
| | VEP | Posterior emissions of EMDA | Posterior emissions of EMDA |
| | CEP1 | Same as CEP | Posterior emissions of EMDA |
| | CEP2 | Posterior emissions of EMS | Same as CEP |

**Specific comments:**

1. Line 40: It should be "Compared with the forecast experiment with prior emission, the forecast with posterior ...". The statement should be revised.

**Response:** Thank you for your comment. We have changed the statement. See lines 40-41, page 2.

"Compared with the forecast with prior emissions, the forecast with posterior emissions significantly improved HCHO simulations, reducing biases by 75.7%, indicating a notable decrease in posterior emission uncertainties.  … …".

2. Line 42: "Moreover" should be deleted. And the statement also should be revised

**Response:** We have deleted the "Moreover" and enhanced the English expression. See lines 43-45, page 2.

"The forecast with posterior emissions also effectively corrected the overestimation of $O_3$ in forecast with prior emissions, reducing biases by 49.3%."

3. Line 176: What did you consider about the boundary condition of NMVOC and NOx?

**Response:** Thank you for this comment. In this study, the boundary conditions for $NO_x$ (including NO and $NO_2$), $O_3$, and HCHO were extracted from the outputs of the Whole Atmosphere Community Climate Model (WACCM). For the other components of NMVOCs, since most NMVOC components have a short atmospheric lifetime (Gaubert et al., 2020; Li et al., 2020). For instance, isoprene, which is the primary component of NMVOCs, has a lifetime of approximately 1 h (Bates and Jacob, 2019). Consequently, the chemical lateral boundary conditions for NMVOCs were just derived from background profiles.

We have added relevant descriptions. See lines 178-183, page 6.

"Chemical lateral boundary conditions for NO, $NO_2$, HCHO, and $O_3$ were extracted from the output of the global CTM (i.e., the Whole Atmosphere Community Climate Model, WACCM) with a resolution of 0.9° × 1.25° at 6-hour intervals (Marsh et al., 2013). Meanwhile, boundary conditions for the other NMVOCs were obtained directly

from background profiles. In the first data assimilation (DA) window, chemical initial conditions (excluding NMVOCs) were also derived from the WACCM outputs, whereas … …"

4.  Line 204~207: Did author consider about the correction of NOx and NMVOCs in the DA system?

**Response:** Yes, NOx and NMVOCs emissions were corrected simultaneously in DA systems. See lines 140-145, page 5.

5.  Line 209~210: As NO2 is a kind of short lifetime gas, the concentration of surface NO2 measurements not only present NO2, but also may include NOx. What did you consider about the influence of NO2 observation uncertain on optimizing NOx emissions?

**Response:** Thank you for this comment. Actually, the perturbed samples of NOx emission in this study are divided to $NO_2$ and NO with a fixed $NO_2$/NO ratio of 1/9 (Zhang et al., 2007). The process of NO being oxidized to $NO_2$ during transport from sources to observation sites is fully taken into account by atmospheric transport models. Therefore, we can directly assimilate $NO_2$ observations to optimize NOx emissions.

6. Line 265: It would be better to use mosaic diagram to present the data amount of TROPOMI HCHO.

**Response:** Thank you for your suggestion. We have used mosaic diagram to present the data amount of TROPOMI HCHO. See Figure 1 in the revised manuscript.

[Figure]

**Figure R1**. Model domain and observation network (a) and data amount of TROPOMI HCHO retrievals during August 2022 in each grid (b). The red dashed frame delineates the CMAQ computational domain; black squares denote surface meteorological measurement sites; navy triangles indicate sounding sites (Text S1), and red and blue dots represent air pollution measurement sites, where red dots are used for assimilation and blue dots for independent evaluation. (Figure 1 in the revised manuscript)

7. Line 299: Please added the year of the study period.

**Response:** Thanks. We have added the year of the study period. See line 326, page 12.

"Before implementing the emission inversion, a relatively perfect initial field is generated at 0000 UTC on August 1, 2022 through conducting a 5-day simulation with 6-hour interval 3D-Var data assimilation."

8. Line 307~314: The background error covariance is implicitly expressed in the EnKF method. How did author implement EMS1 experiment in the DA system? And it would be better to introduce EMS1-3 experiment follow the EMDA, making the text description consistent with the Table1.

**Response:** Thank you for this suggestion. Yes, in the EnKF method, the background error covariance is computed implicitly. However, prior emission uncertainty needs to be provided before implementing the DA system. Specifically, in the EMS1 experiment, we increased the prior uncertainty from 40% to 80%. We have revised this sentence for clarity and precision. See line 313 page 12.

Additionally, we have adjusted the introduction order of those experiments in Section 3, while also removing the EMS1 and EMS2 experiments (See General comments).

"A sensitivity experiment involving a doubling of the prior uncertainty (80%) revealed that the differences in posterior NMVOC emissions… …"

9. Line 324 and 351: "prior and posterior emissions" should be "prior and posterior NMVOC emissions", and "EMGAN" should be "MEGAN".

**Response:** Corrected. Thanks.

See line 364, page 14.

"Figure 2 shows the spatial distribution of temporally averaged prior and posterior NMVOC emissions, along with … …"

See line 401, page 16.

"**Figure 2.** Spatial distribution of the time-averaged (a) prior emissions (MEIC 2020 + MEGAN), (b) posterior emissions … …"

10. Line 440-441, Figure 5: It is difficult for readers to remember the setting of experiments. And I think that "CEP3" should be "CEP1" in the Fig. 5a?

**Response:** Thank you for bringing this oversight to our attention. We have corrected the error. Additionally, we have removed the EMS1, EMS2, and CEP2 experiments from the original manuscript (See General comments).

[Figure]

**Figure R2**. Time series comparison of hourly surface $O_3$ concentrations ($\mu g\ m^{-3}$) and RMSE ($\mu g\ m^{-3}$) from CEP1 and VEP experiments against all observations. (Figure 5 in the revised manuscript)

11. Line 515-518: The background errors and observation errors play an important role in the DA system. It would be better to give a detailed explanation of why the difference in two posterior NMVOC emissions was small by using 'two-step' inversion strategy in the DA system.

**Response:** Thank you for this suggestion. In this study, we innovatively used a "two-step" optimization strategy, in which the emissions are inferred first and then input into the CMAQ model to simulate initial conditions of the next window. That is, the residual error of the current window is reflected in the initial conditions of the next window. Meanwhile, the optimized emissions are transferred to the next window as prior emissions. Therefore, the residual errors from the current assimilation window will be promptly corrected in the next window. This cyclic iteration inversion ensures that the RAPAS system has a relatively low dependence on prior uncertainty settings (Feng et al., 2023). We have included the following discussion in the revised manuscript. Lines 158-161, page 6.

"The inversion process follows a two-step procedure within each inversion window, in which the emissions are inferred first and then input into the CMAQ model to simulate initial conditions of the next window. Meanwhile, the optimized emissions are transferred to the next window as prior emissions. The two-step inversion strategy facilitates error propagation and iterative emission optimization, which have… …"

Lines 314-317, page 12.

"A sensitivity experiment involving a doubling of the prior uncertainty (80%) revealed that the differences in posterior NMVOC emissions amounted to a mere 0.2% (Figure S2). The implementation of a 'two-step' inversion strategy allows for the timely correction of residual errors from the previous assimilation window in the current window, thus ensuring that the RAPAS system has a relatively low dependence on prior uncertainty settings."

**References**

Bates, K.H., Jacob, D.J., 2019. A new model mechanism for atmospheric oxidation of isoprene: global effects on oxidants, nitrogen oxides, organic products, and secondary organic aerosol. Atmos. Chem. Phys. 19, 9613-9640.

Feng, S., Jiang, F., Wu, Z., Wang, H., He, W., Shen, Y., Zhang, L., Zheng, Y., Lou, C., Jiang, Z., Ju, W., 2023. A Regional multi-Air Pollutant Assimilation System (RAPAS v1.0) for emission estimates: system development and application. Geosci. Model Dev. 16, 5949-5977.

Gaubert, B., Emmons, L.K., Raeder, K., Tilmes, S., Miyazaki, K., Arellano Jr, A.F., Elguindi, N., Granier, C., Tang, W., Barré, J., Worden, H.M., Buchholz, R.R., Edwards, D.P., Franke, P., Anderson, J.L., Saunois, M., Schroeder, J., Woo, J.H., Simpson, I.J., Blake, D.R., Meinardi, S., Wennberg, P.O., Crounse, J., Teng, A., Kim, M., Dickerson, R.R., He, H., Ren, X., Pusede, S.E., Diskin, G.S., 2020. Correcting model biases of CO in East Asia: impact on oxidant distributions during KORUS-AQ. Atmos. Chem. Phys. 20, 14617-14647.

Li, C., Li, Q., Tong, D., Wang, Q., Wu, M., Sun, B., Su, G., Tan, L., 2020. Environmental impact and health risk assessment of volatile organic compound emissions during different seasons in Beijing. Journal of Environmental Sciences 93, 1-12.

Zhang, Q., Streets, D.G., He, K., Wang, Y., Richter, A., Burrows, J.P., Uno, I., Jang, C.J., Chen, D., Yao, Z., Lei, Y., 2007. NOx emission trends for China, 1995–2004: The view from the ground and the view from space. Journal of Geophysical Research: Atmospheres 112.

---

## Author Comment (AC2)

**Responses to the comments of Reviewer #1:**

We would like to thank the anonymous referee for his/her comprehensive review and valuable suggestions. These suggestions help us to present our results more clearly. In response, we have made changes according to the referee's suggestions and replied to all comments point by point. All the page and line number for corrections are referred to the revised manuscript, while the page and line number from original reviews are kept intact.

**1.** Accurate NMVOC emissions are essential for predicting air quality. Currently large uncertainties exist in NMVOC emissions, both in the anthropogenic and biogenic sources, as compared to other pollutants, such as SO2 and PM. In this study, the authors use the RAPAS assimilation system incorporated with the EnKF assimilation algorithm to optimize NMVOC emissions using TROPOMI HCHO retrievals. They use MEIC 2020 for anthropogenic emissions and MEGANv2.1 for biogenic sources as the priori NMVOC emissions. They find that NMVOC emissions are largely overestimated, especially biogenic NMVOC emissions. They also find $O_3$ predictions would be lowered using the posterior NMVOC emissions.

The study seems interesting, however, I have a few concerns about some of the key results in this study.

First, the CMAQ model overpredicts $O_3$ in China largely (over 20 ug/m³) in most sites of China (Figure 4a) with the WRF-MEIC-MEGAN setups. The overpredictions are consistent over the whole month (Figure 5). Such 'large' overprediction problem of $O_3$ in CMAQ in China (or any other countries/regions) has not been reported. The overprediction seems very consistent in space and time. The spatial distribution of VOC emissions (in Figure 2a) also looks very uniform. More evaluation and check on the model setups and results should be performed and provided to fully understand this problem. Honestly, attributing such large $O_3$ predictions to VOCs emissions is somewhat dangerous. How are the predictions on CO/ SO2/EC (the species that are less

chemically reactive)? What about the meteorology predictions?

**Response:** We appreciate the reviewer for his/her constructive and up-to-point comments. Actually, it has been observed that in studies involving chemical transport models, there is a tendency for $O_3$ to be overestimated over China. For example, Li et al. (2019) and Akimoto et al. (2019) conducted a model evaluation and intercomparison of surface-level $O_3$ in East Asia in the context of MICS-Asia Phase III. They discovered that 14 state-of-the-art chemical transport models, including the WRF-CMAQ v5.0.2 model with MEIC-MEGAN emissions used in this study, widely overestimated surface $O_3$ over China by 10-30 ppb (20-60 $\mu g/m^3$), especially in the North China Plain and the Pearl River Delta. Liu et al. (2018), Qiao et al. (2019), and Xiong et al. (2023) also found similar $O_3$ overestimation ($\sim$ 20 $\mu g/m^3$) in the Sichuan Basin and the Yangtze River Delta region with the WRF-MEIC-MEGAN setups. Alternatively, in some studies, $O_3$ simulations have been implicitly improved by adjusting model chemical mechanisms or adjusting precursor emissions. For example, to study the effects of emission changes on the worsening of urban ozone pollution in China, Liu and Wang (2020) modified the original CMAQ model to update heterogeneous reactions to weaken the atmospheric oxidation capacity and thus inhibiting $O_3$ formation.

To facilitate the comparison of prior and posterior NMVOC emissions, we used a consistent legend in Figure 2. In fact, although NMVOC emissions are prevalent across much of central and southern China, the higher emissions in Figure 2a are concentrated in places with lush vegetation cover, such as Hunan, Jiangxi, and Zhejiang provinces, as well as in places with intensive anthropogenic activities. Despite our optimization of $O_3$ precursor emissions, the posterior simulations still show some degree of overestimation, indicating the presence of a systematic bias. We agree that the model-data mismatch error not only originates from the emissions, but also from variations in meteorological fields, spatial resolution, model treatments of nonlinear photochemistry and other physical processes. We utilized surface meteorological measurements from 400 stations, including temperature at 2 m (T2), relative humidity at 2 m (RH2), and wind speed at 10 m (WS10), and planetary boundary layer height (PBLH) measured by

sounding from 84 stations to evaluate the performance of WRF simulations (Figure S9 and Text S2). The results showed that the WRF model satisfactorily reproduced T2, RH2, WS10, and PBLH, with small biases of -0.5 °C, -5.3%, 0.3 m/s, and -42.4 m, respectively. The underestimated PBLH may lead to an overestimation of $O_3$, but the overestimated WS10 somewhat compensates for this overestimation. Additionally, due to the relatively coarse spatial resolution, NO titration effects in urban areas may not be well represented in the model, leading to an overestimation of $O_3$ in these areas. Model inherent errors arising from the model structure, parameterization, and the simplification or lack of chemical mechanisms inevitably affect $O_3$ simulations (Li et al., 2020). For example, Li et al. (2018) reported that heterogeneous reactions of nitrogen compounds could reduce surface $O_3$ concentration by 10–20 ppb for the polluted regions over China. These reactions have not been fully incorporated in CMAQ chemical mechanisms. However, there is still a lack of reasonable and effective algorithms to solve the model error in atmospheric data assimilation (Houtekamer and Zhang, 2016).

Due to inconvenient access to EC observation data, we only show the mean bias (BIAS), root mean square error (RMSE), and correlation coefficient (CORR) for simulated CO and $SO_2$ concentrations in the CEP and VEP experiments (Figures R1 and R2). There is a significant underestimation of CO in the CEP with prior emissions. However, a notable underestimation of prior CO emissions (MEIC) of about 100% has been confirmed by inversion estimations (Feng et al., 2020; Tang et al., 2013; Wu et al., 2020) and model evaluations (Kong et al., 2019) in previous studies. The BIAS of $SO_2$ is relatively small. Overall, after optimization, the BIAS and RMSE of CO were reduced from -0.27and 0.36 to -0.09 mg/m$^3$ and 0.21 mg/m$^3$, respectively, and the BIAS and RMSE of $SO_2$ were reduced from -0.36 and 7.0 μg/m$^3$ to -0.35 and 3.34 μg/m$^3$, respectively.

We have added following discussions in the revised manuscript. See lines 545-568, pages 25-26.

"$O_3$ simulations over China have a tendency to be overestimated in studies involving

chemical transport modeling. For example, by intercomparing 14 state-of-the-art CTMs with $O_3$ observations within the framework of the MICS-Asia III, Li et al. (2019) identified a substantial overestimation of annual surface $O_3$ in East Asia, ranging from 20 to 60 μg m$^{-3}$. Notably, the NCP exhibited substantial overestimations, with most models overestimating $O_3$ by 100–200% during May–October. Despite our optimization of $O_3$ precursor emissions, the posterior simulations still exhibit some degree of overestimation (Figure 4), suggesting that there may indeed be an effect of systematic bias, such as meteorological fields, spatial resolution, model treatments of nonlinear photochemistry and other physical processes. The WRF can generally reproduce meteorological conditions sufficiently in terms of their temporal variation and magnitude over China (Figure S9), with small biases of -0.5 °C, -5.3%, 0.3 m/s, and -42.4 m for temperature at 2 m, relative humidity at 2 m, and wind speed at 10 m, and planetary boundary layer height, respectively. However, due to the relatively coarse spatial resolution, NO titration effects in urban areas may not be well represented in the model, which can lead to an overestimation of $O_3$ in these areas. Additionally, model inherent errors arising from the model structure, parameterization, and the simplification or lack of chemical mechanisms inevitably affect the $O_3$ simulations. For example, Li et al. (2018) reported that heterogeneous reactions of nitrogen compounds could weaken the atmospheric oxidation capacity and thus reduce surface $O_3$ concentration by 20–40 μg m$^{-3}$ for the polluted regions over China. These reactions have not been fully incorporated in CMAQ chemical mechanisms. However, there is still a lack of reasonable and effective algorithms for addressing model errors through assimilation (Houtekamer and Zhang, 2016)."

[Figure]

**Figure R1**. Spatial distribution of mean bias (BIAS, a and b), root mean square error (RMSE, c and d), and correlation coefficient (CORR, e and f) for simulated CO using prior (left, CEP) and posterior (right, VEP) emissions, respectively, against observations.

[Figure]

**Figure R2**. Same as Figure R1, but for SO$_2$.

**2.** Figure 4a and Figure 6a seem not consistent, the difference in south China in Figure 6a looks not as significant as the north. Also the observations look no significant spatial variation on Figure 6a, and MDA8 $O_3$ in August is most in blue-green color (~110 ug/m3).

**Response:** Thanks for your comments. Figure 4a shows the BIAS of simulated $O_3$ throughout the entire phase in the CEP1, while Figure 6a shows a comparison between simulated and observed values for MDA8. Figure R3 shows the BIAS of MDA8 simulated in the CEP1 experiment, i.e., simulated minus observed in Figure 6a. A spatial and magnitude resemblance can be observed between Figure R3 and Figure 4a.

The original Figure 6a indeed did not show significant spatial variations. We have readjusted the legend (Figure R4), please refer to Figure 6 in the revised manuscript.

[Figure]

**Figure R3**. Spatial distribution of BIAS for simulated maximum daily 8-hour average (MDA8) $O_3$ concentrations in the CEP1 experiment.

[Figure]

**Figure R4**. Comparisons of (a, b) simulated maximum daily 8-hour average (MDA8) O$_3$ concentrations, (c, d) net reaction rates, (e, f) and differences in production and loss rates between CEP1 and VEP experiments at the surface. Surface MDA8 O$_3$ values (circles) from the national control air quality stations were overlaid (Figure 6 in the revised manuscript)

**3.** What do different symbols/colors in Figure 1 mean?

**Response:** Thanks for this comment. In Figure 1, black squares denote surface meteorological measurement sites; navy triangles indicate sounding sites, and red and blue dots represent air pollution measurement sites, where red dots are used for

assimilation and blue dots for independent evaluation.

We have added a legend in Figure 1, and supplemented the explanation in the caption. See lines 286-287, page 11.

[Figure]

**Figure 1**. Model domain and observation network (a) and data amount of TROPOMI HCHO retrievals during August 2022 in each grid (b). The red dashed frame delineates the CMAQ computational domain; black squares denote surface meteorological measurement sites; navy triangles indicate sounding sites (Text S1), and red and blue dots represent air pollution measurement sites, where red dots are used for assimilation and blue dots for independent evaluation.

**4.** Why not choose 2020 as the study year if you have 2020 MEIC emissions?

**Response:** Thanks for this comment. Yes, we chose 2020 MEIC inventory as the prior emission, and selected August 2022 as the research stage. On one hand, the publicly available MEIC inventory has a lag, currently updated only until 2020. However, our system adopts a 'two-step' inversion strategy, allowing for the timely correction of residual errors from the previous assimilation window in the current window, thus ensuring that the RAPAS system has a relatively low dependence on prior emissions, which has been proven in (Feng et al., 2023). On the other hand, in the summer of 2022, the eastern part of China experienced the strongest and longest-lasting heatwave since 1961 (Wang et al., 2023). High temperatures and drought significantly affect vegetation growth and NMVOC emissions, which also impact $O_3$ production. Such a complex weather system provides a good test for the assimilation capability of our system.

We have added following discussions in the revised manuscript. See lines 321-325, page 12.

"During the summer of 2022, southern China experienced severe heatwave conditions. The combination of high temperatures and drought had a pronounced effect on vegetation growth and NMVOC emissions, thereby influencing $O_3$ production (Wang et al., 2023). Consequently, we opted to focus on August 2022, as it presented an ideal period for testing the assimilation capabilities of our system. Before implementing the emission inversion,… …"

**5.** BVOCs is greatly overestimation by MEGAN (over 50%). Have any other studies reported similar findings with MEGAN in any regions? If no, please explain why such problem occurs in China? Why previous modeling studies in China with CMAQ have not encountered such problems (also the $O_3$ overprediction problem)?

**Response:** Thanks for this comment. The estimation of BVOC emissions by MEGAN is not consistently overestimated. For example, Bauwens et al. (2016) optimized global BVOC emissions using source inversion of OMI HCHO observations and found that

MEGAN tends to overestimate BVOCs in low-latitude regions while underestimating them in high-latitude areas. Research on the inverse estimation of BVOC emissions in China remains limited. For China, Bauwens et al. (2016) showed that the greatest overestimation of BVOC emissions occurred in southern China, up to 45%. The overestimation of BVOC emissions simulated by MEGAN in China is further validated by model evaluation studies (Kim et al., 2017; Kim et al., 2024). In other regions, Marais et al. (2014) found that MEGAN's isoprene emissions were 5-10 times higher than the canopy-scale flux measurements obtained from African field campaigns in equatorial forest and woody savannas. Warneke et al. (2010) found that MEGAN overestimated BVOC emissions by up to a factor of two when compared to estimates based on airborne measurements over Texas. Millet et al. (2008) compared isoprene emissions derived using satellite-observed HCHO columns with MEGAN emissions for North America, noting an average overestimation of BVOC emissions by a factor of 2, reaching up to a factor of 5 in certain locations. Similarly, Wang et al. (2017) observed a significant overestimation of BVOC emissions by MEGAN, averaging a factor of 3 in the United States. Kaiser et al. (2018) applied an adjoint algorithm to estimate isoprene emission over the southeast US, revealing an average overestimation of MEGAN-derived BVOC emissions by 40%, slightly lower than the 50% overestimations reported by Bauwens et al. (2016). Additionally, Chaliyakunnel et al. (2019) found the modeled BVOC emissions using MEGAN were overestimated by approximately 30–60% for most locations and seasons. Therefore, there is indeed a possibility of significant uncertainty in MEGAN.

The significant decrease in BVOC emissions observed in this study may also be influenced by other factors. Apart from inaccuracies in the distribution of plant functional types, empirical parameterization, especially concerning responses to temperature and drought stress, can introduce substantial uncertainties (Angot et al., 2020; Jiang et al., 2018). Zhang et al. (2021) highlighted that the temperature-dependent activity factor increases evidently with rising temperatures in the MEGAN model. Additionally, Wang et al. (2021) pointed out that the missing of a drought scheme is

one of the factors causing the isoprene overestimation in the MEGAN model. Wang et al. (2022) applied new drought stress algorithms to simulate the impact of drought on isoprene emission and found that drought can decrease isoprene emission globally by 11%. During the summer of 2022, southern China experienced severe heatwave conditions. The MEGAN model may not effectively capture the impacts of high temperatures and drought on vegetation, leading to significant uncertainties in BVOC emissions.

It has been widely observed in existing studies that there is a trend of overestimation of $O_3$ in China, which is similar to the overestimation found in this study. Please refer to Comment 1 for further details.

We have added the following discussions. See lines 374-390, page 15.

"Additionally, uncertainties in MEGAN parameterization have significant implications for NMVOC emission estimations, particularly concerning the responses of vegetation in MEGAN to temperature and drought stress (Angot et al., 2020; Jiang et al., 2018). Zhang et al. (2021) highlighted that the temperature-dependent activity factor noticeably increases with rising temperatures in MEGAN. Wang et al. (2021b) pointed out that the missing of a drought scheme is one of the factors causing the overestimation of isoprene emissions in MEGAN. Opacka et al. (2022) optimized the empirical parameter in the MEGANv2.1 soil moisture stress algorithm, resulting in significant reductions in isoprene emissions and providing better agreement between modelled and observed HCHO temporal variability in the central U.S. During the study period, China experienced severe heatwave conditions, which may further hinder the MEGAN's ability to effectively capture the impacts of high temperatures and drought on vegetation, thus resulting in significant overestimation in NMVOC emissions (Wang et al., 2022). Nevertheless, the large magnitude of emission reductions of 50.2% in our inversion is comparable to studies in southern China (Bauwens et al., 2016; Zhou et al., 2023), southeastern US (Kaiser et al., 2018), Africa (Marais et al., 2014), India (Chaliyakunnel et al., 2019), Amazonia (Bauwens et al., 2016), and parts of Europe (Curci et al., 2010), but opposite to the large-scale emission increase over China in Cao et al. (2018). For $NO_x$ (Figure S4), … …"

**References**

Akimoto, H., Nagashima, T., Li, J., Fu, J.S., Ji, D., Tan, J., Wang, Z., 2019. Comparison of surface ozone simulation among selected regional models in MICS-Asia III – effects of chemistry and vertical transport for the causes of difference. Atmos. Chem. Phys. 19, 603-615.

Angot, H., McErlean, K., Hu, L., Millet, D.B., Hueber, J., Cui, K., Moss, J., Wielgasz, C., Milligan, T., Ketcherside, D., Bret-Harte, M.S., Helmig, D., 2020. Biogenic volatile organic compound ambient mixing ratios and emission rates in the Alaskan Arctic tundra. Biogeosciences 17, 6219-6236.

Bauwens, M., Stavrakou, T., Müller, J.F., De Smedt, I., Van Roozendael, M., van der Werf, G.R., Wiedinmyer, C., Kaiser, J.W., Sindelarova, K., Guenther, A., 2016. Nine years of global hydrocarbon emissions based on source inversion of OMI formaldehyde observations. Atmos. Chem. Phys. 16, 10133-10158.

Chaliyakunnel, S., Millet, D.B., Chen, X., 2019. Constraining Emissions of Volatile Organic Compounds Over the Indian Subcontinent Using Space-Based Formaldehyde Measurements. Journal of Geophysical Research: Atmospheres 124, 10525-10545.

Feng, S., Jiang, F., Wu, Z., Wang, H., He, W., Shen, Y., Zhang, L., Zheng, Y., Lou, C., Jiang, Z., Ju, W., 2023. A Regional multi-Air Pollutant Assimilation System (RAPAS v1.0) for emission estimates: system development and application. Geosci. Model Dev. 16, 5949-5977.

Feng, S., Jiang, F., Wu, Z., Wang, H., Ju, W., Wang, H., 2020. CO Emissions Inferred From Surface CO Observations Over China in December 2013 and 2017. Journal of Geophysical Research-Atmospheres 125.

Houtekamer, P.L., Zhang, F., 2016. Review of the Ensemble Kalman Filter for Atmospheric Data Assimilation. Monthly Weather Review 144, 4489-4532.

Jiang, X., Guenther, A., Potosnak, M., Geron, C., Seco, R., Karl, T., Kim, S., Gu, L., Pallardy, S., 2018. Isoprene emission response to drought and the impact on global atmospheric chemistry. Atmospheric Environment 183, 69-83.

Kaiser, J., Jacob, D.J., Zhu, L., Travis, K.R., Fisher, J.A., González Abad, G., Zhang, L., Zhang, X., Fried, A., Crounse, J.D., St. Clair, J.M., Wisthaler, A., 2018. High-resolution inversion of OMI formaldehyde columns to quantify isoprene emission on ecosystem-relevant scales: application to the southeast US. Atmos. Chem. Phys. 18, 5483-5497.

Kim, E., Kim, B.-U., Kim, H.C., Kim, S., 2017. The Variability of Ozone Sensitivity to Anthropogenic Emissions with Biogenic Emissions Modeled by MEGAN and BEIS3.

Atmosphere 8, 187.

Kim, K.M., Kim, S.W., Seo, S., Blake, D.R., Cho, S., Crawford, J.H., Emmons, L.K., Fried, A., Herman, J.R., Hong, J., Jung, J., Pfister, G.G., Weinheimer, A.J., Woo, J.H., Zhang, Q., 2024. Sensitivity of the WRF-Chem v4.4 simulations of ozone and formaldehyde and their precursors to multiple bottom-up emission inventories over East Asia during the KORUS-AQ 2016 field campaign. Geosci. Model Dev. 17, 1931-1955.

Kong, L., Tang, X., Zhu, J., Wang, Z., Fu, J.S., Wang, X., Itahashi, S., Yamaji, K., Nagashima, T., Lee, H.J., Kim, C.H., Lin, C.Y., Chen, L., Zhang, M., Tao, Z., Li, J., Kajino, M., Liao, H., Sudo, K., Wang, Y., Pan, Y., Tang, G., Li, M., Wu, Q., Ge, B., Carmichael, G.R., 2019. Evaluation and uncertainty investigation of the NO2, CO and NH3 modeling over China under the framework of MICS-Asia III. Atmos. Chem. Phys. Discuss. 2019, 1-33.

Li, J., Chen, X., Wang, Z., Du, H., Yang, W., Sun, Y., Hu, B., Li, J., Wang, W., Wang, T., Fu, P., Huang, H., 2018. Radiative and heterogeneous chemical effects of aerosols on ozone and inorganic aerosols over East Asia. Science of The Total Environment 622-623, 1327-1342.

Li, J., Nagashima, T., Kong, L., Ge, B., Yamaji, K., Fu, J.S., Wang, X., Fan, Q., Itahashi, S., Lee, H.J., Kim, C.H., Lin, C.Y., Zhang, M., Tao, Z., Kajino, M., Liao, H., Li, M., Woo, J.H., Kurokawa, J., Wang, Z., Wu, Q., Akimoto, H., Carmichael, G.R., Wang, Z., 2019. Model evaluation and intercomparison of surface-level ozone and relevant species in East Asia in the context of MICS-Asia Phase III – Part 1: Overview. Atmos. Chem. Phys. 19, 12993-13015.

Li, J., Zhang, H., Ying, Q., Wu, Z., Zhang, Y., Wang, X., Li, X., Sun, Y., Hu, M., Zhang, Y., Hu, J., 2020. Impacts of water partitioning and polarity of organic compounds on secondary organic aerosol over eastern China. Atmos. Chem. Phys. 20, 7291-7306.

Liu, Y., Li, L., An, J., Huang, L., Yan, R., Huang, C., Wang, H., Wang, Q., Wang, M., Zhang, W., 2018. Estimation of biogenic VOC emissions and its impact on ozone formation over the Yangtze River Delta region, China. Atmospheric Environment 186, 113-128.

Liu, Y., Wang, T., 2020. Worsening urban ozone pollution in China from 2013 to 2017 – Part 2: The effects of emission changes and implications for multi-pollutant control. Atmos. Chem. Phys. 20, 6323-6337.

Marais, E.A., Jacob, D.J., Guenther, A., Chance, K., Kurosu, T.P., Murphy, J.G., Reeves, C.E., Pye, H.O.T., 2014. Improved model of isoprene emissions in Africa using Ozone Monitoring Instrument (OMI) satellite observations of formaldehyde: implications for oxidants and particulate matter. Atmos. Chem. Phys. 14, 7693-7703.

Millet, D.B., Jacob, D.J., Boersma, K.F., Fu, T.-M., Kurosu, T.P., Chance, K., Heald, C.L., Guenther, A., 2008. Spatial distribution of isoprene emissions from North America derived from formaldehyde column measurements by the OMI satellite sensor. Journal of Geophysical Research: Atmospheres 113.

Qiao, X., Guo, H., Wang, P., Tang, Y., Ying, Q., Zhao, X., Deng, W., Zhang, H., 2019. Fine Particulate Matter and Ozone Pollution in the 18 Cities of the Sichuan Basin in Southwestern China: Model Performance and Characteristics. Aerosol and Air Quality Research 19, 2308-2319.

Tang, X., Zhu, J., Wang, Z.F., Wang, M., Gbaguidi, A., Li, J., Shao, M., Tang, G.Q., Ji, D.S., 2013. Inversion of CO emissions over Beijing and its surrounding areas with ensemble Kalman filter. Atmospheric Environment 81, 676-686.

Wang, H., Lu, X., Seco, R., Stavrakou, T., Karl, T., Jiang, X., Gu, L., Guenther, A.B., 2022. Modeling Isoprene Emission Response to Drought and Heatwaves Within MEGAN Using Evapotranspiration Data and by Coupling With the Community Land Model. Journal of Advances in Modeling Earth Systems 14, e2022MS003174.

Wang, J., Yan, R., Wu, G., Liu, Y., Wang, M., Zeng, N., Jiang, F., Wang, H., He, W., Wu, M., Ju, W., Chen, J.M., 2023. Unprecedented decline in photosynthesis caused by summer 2022 record-breaking compound drought-heatwave over Yangtze River Basin. Science Bulletin 68, 2160-2163.

Wang, P., Liu, Y., Dai, J., Fu, X., Wang, X., Guenther, A., Wang, T., 2021. Isoprene Emissions Response to Drought and the Impacts on Ozone and SOA in China. Journal of Geophysical Research: Atmospheres 126, e2020JD033263.

Wang, P., Schade, G., Estes, M., Ying, Q., 2017. Improved MEGAN predictions of biogenic isoprene in the contiguous United States. Atmospheric Environment 148, 337-351.

Warneke, C., de Gouw, J.A., Del Negro, L., Brioude, J., McKeen, S., Stark, H., Kuster, W.C., Goldan, P.D., Trainer, M., Fehsenfeld, F.C., Wiedinmyer, C., Guenther, A.B., Hansel, A., Wisthaler, A., Atlas, E., Holloway, J.S., Ryerson, T.B., Peischl, J., Huey, L.G., Hanks, A.T.C., 2010. Biogenic emission measurement and inventories determination of biogenic emissions in the eastern United States and Texas and comparison with biogenic emission inventories. Journal of Geophysical Research: Atmospheres 115.

Wu, H., Tang, X., Wang, Z., Wu, L., Li, J., Wang, W., Yang, W., Zhu, J., 2020. High-spatiotemporal-resolution inverse estimation of CO and NOx emission reductions during emission control periods with a modified ensemble Kalman filter. Atmospheric Environment 236.

Xiong, K., Xie, X., Mao, J., Wang, K., Huang, L., Li, J., Hu, J., 2023. Improving the accuracy of O3 prediction from a chemical transport model with a random forest model in the Yangtze River Delta region, China. Environmental Pollution 319, 120926.

Zhang, M., Zhao, C., Yang, Y., Du, Q., Shen, Y., Lin, S., Gu, D., Su, W., Liu, C., 2021. Modeling sensitivities of BVOCs to different versions of MEGAN emission schemes in WRF-Chem (v3.6) and its impacts over eastern China. Geosci. Model Dev. 14, 6155-6175.

---

## Author Response (AR2)

**Responses to the comments of Reviewer #1:**

We would like to thank the anonymous referee for his/her comprehensive review and valuable suggestions. These suggestions help us to present our results more clearly. In response, we have made changes according to the referee's suggestions and replied to all comments point by point.

1.  the conclusion about MEGAN overestimates NMVOC emissions seems to be case-specific, and in the revision the authors emphasize that it is due to severe heatwave conditions. So I would suggest that 'August 2022' should be clearly stated in the title and the abstract. In the discussion, it is also suggested to mention that the conclusion is for august 2022 case.

**Response:** Thanks for your suggestions. We have added "August 2022" to the title and modified the title to " Constraining Non-Methane VOC Emissions with TROPOMI HCHO observations: Impact on Summertime Ozone Simulation in August 2022 in China".

We also added corresponding statements to the abstract and conclusion

See line 33, page 2.

"Here, we extended the Regional multi-Air Pollutant Assimilation System (RAPAS) with the EnKF algorithm to optimize NMVOC emissions in China in August 2022 by assimilating TROPOMI HCHO retrievals."

See lines 38-41, page 2.

"NMVOC emissions exhibited a substantial reduction of 50.2%, especially in the middle and lower reaches of the Yangtze River, revealing a prior overestimation of biogenic NMVOC emissions due to extreme heatwave."

See line 592, page 26.

"The application of TROPOMI HCHO observations as constraints led to a substantial

reduction of 50.2% compared to the prior emissions for NMVOCs in August 2022."

2. The data of Figure 5 should be more clearly stated in the text. Average across how many sites? It would be good to illustrate in a few key sites/cities and put in SI, so that the readers can get more detailed information.

**Response:** Thanks for your suggestions. We have provided detailed statistics and the number of sites in the text. Additionally, we conducted statistics on seven key cities selected from the seven major regions constituting China. The detailed changes are shown below:

See lines 457-462, pages 18-19.

"Constraining the NMVOC emissions also led to better model simulations in terms of RMSE throughout the entire study period. Time-averaged BIAS and RMSE decreased from 20.6 and 37.3 $\mu$g m$^{-3}$ to 10.6 and 31.0 $\mu$g m$^{-3}$, respectively. We also evaluated the simulation results for seven key cities (i.e., Beijing, Shanghai, Guangzhou, Wuhan, Chongqing, Yinchuan, and Changchun, which represent key cities in North, East, South, Central, Southwest, Northwest, and Northeast China, respectively), and the biases in the VEP with posterior emissions all showed a significant reduction (Figure S8). Overall, the assimilation of HCHO column observations effectively reduced NMVOC emission uncertainties and consequently improved simulations of HCHO and $O_3$."

[Figure]

**Figure 5**. Time series comparison of hourly surface $O_3$ concentrations ($\mu g\ m^{-3}$) and RMSE ($\mu g\ m^{-3}$) from CEP1 and VEP experiments against all observations at 1701 monitoring sites. The blue and red values on the graph represent the time-averaged statistics in the CEP1 and VEP experiments, respectively.

See Figure S8 in the Supplementary Information.

[Figure]

**Figure S8**. Time series comparison of simulated and observed hourly surface $O_3$ concentrations ($\mu g\ m^{-3}$) from CEP1 and VEP experiments over (a) Beijing, (b) Shanghai, (c) Guangzhou, (d) Wuhan, (e) Chongqing, (f) Yinchuan, and (g) Changchun, representing key cities in North China, East China, South China, Central China, Southwest China, Northwest China, and Northeast China, respectively.

3.   Espeically in urban areas, where anthropogenic emissions become dominant for VOCs, are the anthropogenic NMVOCs emissions also overestimated? How is it compared to rural areas where biogenic emissions dominate?

**Response:** Thanks for this comment. Accurately distinguishing between biogenic and anthropogenic NMVOC emissions remains challenging. This difficulty arises because the same model grid contains both biogenic and anthropogenic emissions, and observations cannot easily differentiate the mixed signals in the atmosphere. In this study, we derived the posterior anthropogenic emissions based on the prior emission ratio information and the total posterior emissions (Figure R1). It can be observed that in most regions of the central and eastern parts of China, especially in areas where anthropogenic sources absolutely dominate the total emissions, such as Beijing-Tianjin-Hebei, the Yangtze River Delta, and the Pearl River Delta, there is a significant decrease in posterior NMVOC emissions. The overall NMVOC emissions in China have decreased by 43.4%, but this is within the uncertainty range of MEIC (±68-78%) (Li et al., 2019). Additionally, the overestimation of anthropogenic emissions in urban areas is less than that of biogenic emissions in rural areas (53.7%).

We have added following descriptions in the revised manuscript. See lines 371-374, page 13.

"Ultimately, the biogenic NMVOC emissions decreased by 53.7%, which was higher than the 43.4% decrease in anthropogenic NMVOC emissions (Figure S3). Overall, the large magnitude of emission decrease of 50.2% in our inversion is comparable to studies … …"

[Figure]

Figure R1. Spatial distribution of the time-averaged (a) prior anthropogenic emissions (MEIC 2020), (b) posterior anthropogenic emissions, (c) difference between prior and posterior anthropogenic emissions (posterior minus prior), (d) prior biogenic emissions (MEGAN), (e) posterior biogenic emissions, (f) difference between prior and posterior biogenic emissions (posterior minus prior) of NMVOCs over China. (Figure S3 in the Supplementary Information)

Li, M., Zhang, Q., Zheng, B., Tong, D., Lei, Y., Liu, F., Hong, C., Kang, S., Yan, L., Zhang, Y., Bo, Y., Su, H., Cheng, Y., and He, K.: Persistent growth of anthropogenic non-methane volatile organic compound (NMVOC) emissions in China during 1990–2017: drivers, speciation and ozone formation potential, Atmos. Chem. Phys., 19, 8897-8913, 10.5194/acp-19-8897-2019, 2019.